# Ternary aromatic and anti-aromatic clusters derived from the *hypho* species [Sn$_2$Sb$_5$]$^{3-}$

Yu-He Xu[1], Nikolay V. Tkachenko [2], Ivan A. Popov [3], Lei Qiao [1], Alvaro Muñoz-Castro[4], Alexander I. Boldyrev [2] & Zhong-Ming Sun [1✉]

Heterometallic clusters have attracted broad interests in the synthetic chemistry due to their various coordination modes and potential applications in heterogeneous catalysis. Here we report the synthesis, experimental, and theoretical characterizations of four ternary clusters ([M$_2$(CO)$_6$Sn$_2$Sb$_5$]$^{3-}$ (M = Cr, Mo), and [(MSn$_2$Sb$_5$)$_2$]$^{4-}$, (M = Cu, Ag)) in the process of capturing the *hypho*- [Sn$_2$Sb$_5$]$^{3-}$ in ethylenediamine (en) solution. We show that the coordination of the binary anion to transition-metal ions or fragments provides additional stabilization due to the formation of locally σ-aromatic units, producing a spherical aromatic shielding region in the cages. While in the case of [Mo$_2$(CO)$_6$Sn$_2$Sb$_5$]$^{3-}$ stabilization arises from locally σ-aromatic three-centre and five-centre two-electron bonds, aromatic islands in [(AgSn$_2$Sb$_5$)$_2$]$^{4-}$ and [(CuSn$_2$Sb$_5$)$_2$]$^{4-}$ render them globally antiaromatic. This work describes the coordination chemistry of the versatile building block [Sn$_2$Sb$_5$]$^{3-}$, thus providing conceptual advances in the field of metal-metal bonding in clusters.

[1] State Key Laboratory of Elemento-Organic Chemistry, Tianjin Key Lab for Rare Earth Materials and Applications, School of Materials Science and Engineering, Nankai University, Tianjin, China. [2] Department of Chemistry and Biochemistry, Utah State University, Logan, UT, USA. [3] Theoretical Division, Los Alamos National Laboratory, Los Alamos, NM, USA. [4] Grupo de Química Inorgánicay Materiales Moleculares, Facultad de Ingenieria, Universidad Autonoma de Chile, El Llano Subercaseaux, Santiago, Chile. ✉email: sunlab@nankai.edu.cn

Zintl-precursors play a key role in the construction of numerous metal clusters via complex coordination. The first structural determination of a Zintl anion was made by Kummer and Diehl in the 1970s, when they revealed the presence of $[Sn_9]^{4-}$ cluster, which was subsequently used as a precursor for the synthesis of many polystannides[1]. Afterward, many more homo- and heteroatomic polyanions were discovered including $[E_9]^{4-}$ (E = Si-Pb), $[Pn_7]^{3-}$ (Pn = P-Bi) and $[Tt_2Pn_2]^{2-}$ (Tt = Ge-Pb)[2–11], and a number of new clusters were produced using them as precursors[12–15]. In the majority of these clusters, the relationship between structure, bonding, and valence electron count can be understood in terms of either the Zintl-Klemm concept addressing electron-precise structures composed of formal two-center two-electron (2c-2e)bonds[16,17] or the Wade-Mingos rules[18–20], which were originally applied to electron-deficient boranes with delocalized bonding, and later introduced into Zintl clusters. According to Wade-Mingos rules, $[B_{10}H_{10}]^{2-}$ is a classic *closo*-type polyhedron (Fig. 1a), isostructural with $[Ge_{10}]^{2-}$ and $[Pb_{10}]^{2-}$. The most commonly used Zintl-precursors $[E_9]^{4-}$ (E = Si-Pb) of the *nido*- type can be formed by removing one 4-connected vertex from the *closo*-type (Fig. 1b)[21,22]. Lately, Dehnen and coworkers reported the first *arachno*-Zintl ion $[Sn_5Sb_3]^{3-}$, which fills the gap in the series of the known compounds (Fig. 1c)[23]. The only reported *hypho*-Zintl anion, $[Sn_3Bi_3]^{5-}$ with high negative charges, was obtained only in liquid ammonia[24]. However, the *hypho*-cage derived from the 10-vertex $[B_{10}H_{10}]^{2-}$ *closo*-type has not yet been observed.

Recently, we adopted self-assembly as a synthetic strategy to synthesize a series of large Au/Pb clusters $[Au_8Pb_{33}]^{6-}$, $[Au_{12}Pb_{44}]^{8-}$, constructed from a *nido*-type cluster $[AuPb_{11}]^{3-}$[25]. As part of this ongoing study, we aim to build metal clusters using other unnoticed Zintl precursors and the solid-state Zintl phase "$K_8SnSb_4$" came into our sight[26]. Though no new precursor has yet been isolated from its solution, a family of clusters all containing a quadricyclane-like cage $[Sn_2Sb_5]^{3-}$ are crystallized in the reaction of $K_8SnSb_4$ with oxidizing organometallic compounds (Fig. 1d). Herein, we report the syntheses and characterizations of these clusters $[M_2(CO)_6Sn_2Sb_5]^{3-}$ (M = Cr, **1**; Mo, **2**), $[(MSn_2Sb_5)_2]^{4-}$ (M = Ag, **3**; Cu, **4**). This series corresponds to a vertex supplement process, from *hypho*- to *arachno*-, and then to *nido*- type. Our analysis of the electronic structure also indicates that while clusters **1** and **2** are aromatic, **3** and **4** feature anti-aromaticity. It is also found that the solid phase "$K_8SnSb_4$" could act as a versatile precursor to construct multinary cluster anions, and $[Sn_2Sb_5]^{3-}$ may serve as a potential building unit for further reactions.

## Results

### Synthesis and characterization.
An en solution of $K_8SnSb_4$ reacted with $Cr(CO)_6$ or $Mo(CO)_6$ in the presence of [2.2.2]-crypt to give rise to *nido*-type clusters $[Cr_2(CO)_6Sn_2Sb_5]^{3-}$ (**1**) or $[Mo_2(CO)_6Sn_2Sb_5]^{3-}$ (**2**). The compounds [K(2.2.2-crypt)]$_4$[(AgSn$_2$Sb$_5$)$_2$] (**3′**) and [K(2.2.2-crypt)]$_4$[(CuSn$_2$Sb$_5$)$_2$] (**4′**)

were obtained from the reaction of $K_8SnSb_4$ with $Ag_4(Mes)_4$ and $Cu(PPh_3)Cl$, respectively. Fig. 2a summarizes the reactions reported here. The ESI mass spectrum (Fig. 2b, Supplementary Figs. 10–25) combined with EDX data (Supplementary Figs. 26–29) confirms a 2:5 Sn:Sb ratio in all four clusters, while X-ray diffraction confirmed the positions of the heavy atoms in the clusters. The overall charges on the clusters (3− on **1** and **2**, 4− on **3** and **4**), in conjunction with the typical charges on the transition metal fragments ($[Cr/Mo(CO)_3]^0$, $Cu^+/Ag^+$), suggests that the $Sn_2Sb_5$ fragment is at the formal 3− charge state. To verify that, our structural studies were complemented by DFT calculations on the isolated $[Sn_2Sb_5]^{3-}$ unit. Various isomeric structures of the $[Sn_2Sb_5]^{3-}$ stoichiometry were explored via the unbiased Coalescence Kick[27,28] (CK) algorithm and their total energies were compared (the lowest energy metastable isomers are shown in Supplementary Table 6). Three low-lying isomers exhibit cognate geometrical structures, differing in the positions of the Sb and Sn atoms (Supplementary Table 6, Supplementary Data 1). These isomers can be considered energetically indistinguishable at the DFT level as the energy difference was calculated to be within ~3 kcal/mol by most of the functionals. By comparing with the experimental X-ray data, it is evident that the isomer $C_{2v}$-$[Sn_2Sb_5]^{3-}$ describes well all structural features obtained from the experiment, better matching the atomic positions than the other two less symmetrical isomers (Supplementary Fig. 43). It is worthy to note that the described tri-anion $[Sn_2Sb_5]^{3-}$ species were not expected to be thermodynamically stable toward autodetachment of an electron, or structure dissociation. However, it was further proved that the results obtained for the $[Sn_2Sb_5]^{3-}$ species are similar to those of the neutral $K_3[Sn_2Sb_5]$ complexes, which are stable towards electrons ejections and structural degradation[29]. The CK calculations for $K_3[Sn_2Sb_5]$ confirmed similar geometrical features of the $Sn_2Sb_5$ core with the three low-lying isomers including $C_{2v}$-$K_3[Sn_2Sb_5]$ (Supplementary Table 7, Supplementary Data 1). The energy difference between these isomers becomes even smaller, i.e., ~1 kcal/mol at most of the DFT functionals, and ~1.7 kcal/mol at the more accurate CCSD level, thus confirming that they are indeed energetically degenerate.

It is important to note that silver and copper mirrors were observed on the wall of the test tubes during the synthesis of **3** and **4**, indicating that $Ag^I$ and $Cu^I$ are acting as oxidizing agents. However, we have not been able to isolate naked $[Sn_2Sb_5]^{3-}$ (**5**) that may be indicative of the high degree of strain and that this cluster can be stabilized only through coordination to transition metal ions. In this sense, it is very similar to the case of the ozone-like $[Bi_3]^{3-}$ cluster[30].

As shown in Fig. 3, all four clusters contain the same *hypho*-structure unit $[Sn_2Sb_5]^{3-}$ (**5**) constructed by two 3-atom and 5-atom faces and one adjacent 4-atom face, which can be

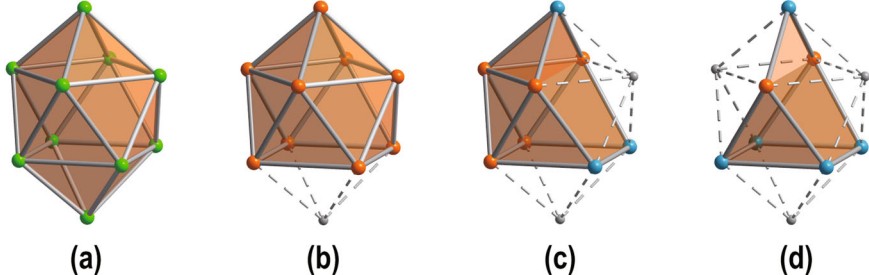

**Fig. 1 Four representative structures from *closo* to *hypho*-type (this work). a** $[B_{10}H_{10}]^{2-}$, **b** $[E_9]^{4-}$ (E = Si-Pb), **c** $[Sn_5Sb_3]^{3-}$, **d** *hypho*-$[Sn_2Sb_5]^{3-}$. The globe colors are as follow: B = green, Tt (Si-Pb) = orange, Sb = blue.

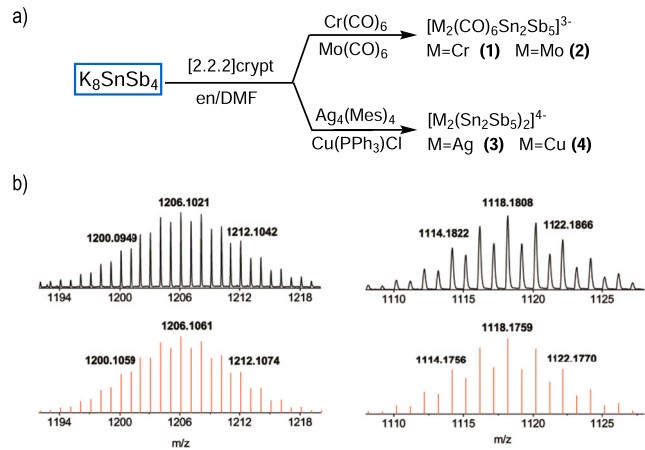

| Table 1 Comparison of selected atomic distances for clusters 1, 2, 3, and 4. | | | | |
|---|---|---|---|---|
| | **distance/Å** | | | |
| Cluster | Sb–Sb | Sn–Sb | M–Sn | M–Sb1/Sb1' |
| 1 | 2.785–2.948 | 2.784–2.934 | 3.125–3.203 | |
| 2 | 2.803–2.926 | 2.813–2.967 | 3.179–3.241 | |
| 3 | 2.790–2.930 | 2.790–2.945 | 2.970–3.025 | 2.917/2.758 |
| 4 | 2.816–2.942 | 2.816–2.913 | 2.815–2.860 | 2.669/2.616 |

**Fig. 2 Reaction diagram and major ESI (−) mass spectra. a** Reaction diagram for the synthesis of clusters **1–4**. **b** ESI (−) mass peaks corresponding to $[Cr_2(CO)_6Sn_2Sb_5]^-$ (left) and $[Mo_2(CO)_6Sn_2Sb_5]^-$ (right). The experimental mass distributions are depicted in black, and the theoretical masses of the isotope distributions are shown in red (the experimental values are labeled to the fourth digit).

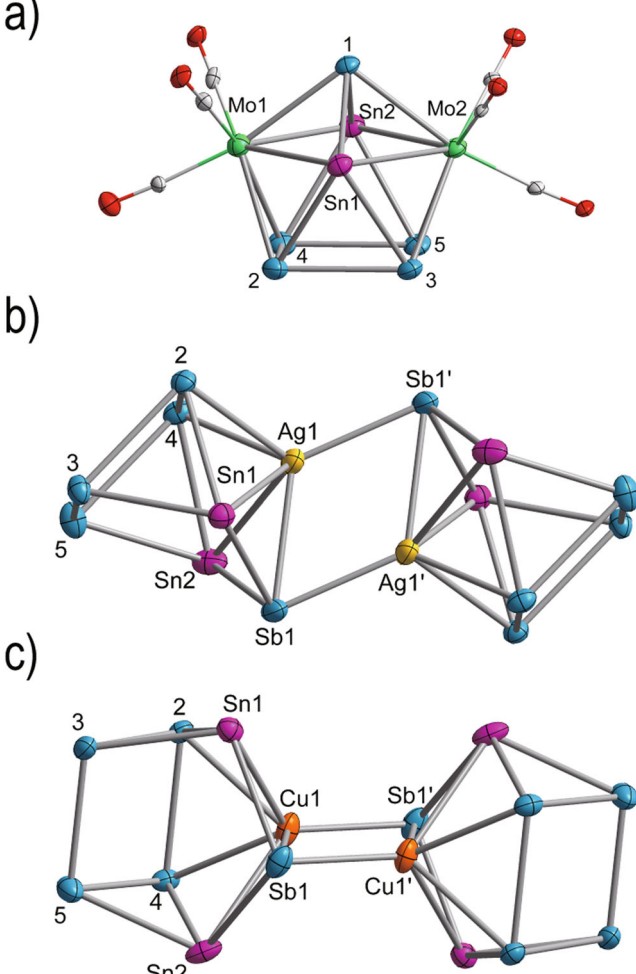

**Fig. 3 Ellipsoid plot (50% level) of the crystal structures. a** $[Mo_2(CO)_6Sn_2Sb_5]^{3-}$, **b** $[(AgSn_2Sb_5)_2]^{4-}$, **c** $[(CuSn_2Sb_5)_2]^{4-}$ (**b, c** represent front and side views of the same structural type). The ellipsoid colors are as follow: Sn = pink, Sb = blue, Mo = green, Ag = yellow, Cu = orange, C = gray and O = red.

regarded as derived from the 10-vertex *closo* deltahedra by removing one 4-connected and two 5-connected vertices. Clusters **1** and **2** are isostructural, and contain nine atoms in a distorted mono-capped square antiprismatic geometry, with Sb atoms occupying all five 4-connected vertices and two Sn and Mo atoms at the 5-connected vertices. The two zero-electron donor transition metal fragments $Mo(CO)_3$ coordinate to the 5-membered rings of the $Sn_2Sb_5$ cluster and form an expanded square plane $Mo_2Sn_2$ (3.1788(8)–3.2407(8) Å) compared with the bottom $Sb_4$ plane (2.8031(9)–2.9259(8) Å). All Mo–Sb bond lengths fall in a narrow range, from 2.9250(8) Å to 2.9538(8) Å, comparable to those reported in the $[Sb_7Mo(CO)_3]^{3-}$ anion (2.942(8)–2.951(9)) Å[31].

The anionic clusters $[(AgSn_2Sb_5)_2]^{4-}$ and $[(CuSn_2Sb_5)_2]^{4-}$ are also isostructural and can be described as a dimer of two *arachno*-cages $(MSn_2Sb_5)^{2-}$ sharing an $M_2Sb_2$ diamond plane. From the structural perspective, the *hypho*-$[Sn_2Sb_5]^{3-}$ is very similar to that in **1** and **2** and the comparison of selected bond distances are listed in Table 1. By introducing a $Cu^+$ or $Ag^+$ ion to occupy one of the above-mentioned 5-connected vertices, **5** is transformed to an *arachno*-cage $(MSn_2Sb_5)^{2-}$, which is isostructural with the recently reported Zintl ion $[Sn_5Sb_3]^{3-}$ [23]. The two $(MSn_2Sb_5)^{2-}$ cages are arranged upside down resulting in two parallel $Sb_4$ planes. In the Ag system (**3**), the Ag–Sn distances range from 2.970(3) Å to 3.025(3) Å, similar to those in $[Ag(Sn_9-Sn_9)]^{5-}$ (2.880(2)–3.010(1) Å)[32], but longer than the typical Ag–Sn single bonds. There are limited opportunities to compare the Ag–Sb bonds with literature precedents because only a few organometallic compounds such as $[Ag_{12}\{Sb(SiMe_3)\}_6(^{Pi}Pr_3)_6]$ (2.722 (10)–2.746(12) Å) are available as ref. [33]. There are three types of Ag–Sb bond lengths in cluster **3**. Ag1–Sb1–Ag1'–Sb1' construct a diamond plane, in which Ag1–Sb1' and Ag1'–Sb1 are both 2.758(14) Å, similar to most Ag–Sb single bonds. However, Ag1–Sb1 and Ag1'–Sb1' are 2.917(15) Å, significantly longer than those of the reported complexes. The remaining Ag–Sb bonds lie in a narrow range of 2.874(2)–2.875(3) Å. The only Ag–Ag bond is 2.802(2) Å, shorter than the sum of covalent radii for a single Ag–Ag bond, i.e., 2.88 Å. However, the $Cu_2Sb_2$ diamond plane is compressed compared to $Ag_2Sb_2$ in **4**. The bond length of Cu–Sb1 (2.669 Å) is akin to Cu–Sb1' (2.616 Å), close to those of $\{[CuSn_5Sb_3]^{2-}\}_2$ (2.619(2)–2.641(2) Å)[34], but considerably longer than typical Cu–Sb single bond, 2.554(9) Å in $[ClCu(SbPh_3)_3]$[35]. The Cu–Cu distance is 2.545 Å, well comparable with those of $[Cu_4@Sn_{18}]^{4-}$ (2.5292(12)−2.5511(13) Å) and $\{[CuGe_9Mes]_2\}^{4-}$ (2.5214(7) Å)[36,37].

**Structure and chemical bonding.** According to Wade-Mingos and *mno* rules[18–20,38], every naked Sn atom bearing a lone pair provides two electrons for cluster bonding ($2 \times 2$), and each Sb atom owns three skeleton electrons ($3 \times 5$), plus three negative charges, leading to a total of 22 skeleton electrons ($2n + 8$). The cluster contains one polyhedron ($m = 1$), seven atoms ($n = 7$) and three missing vertices ($o = 3$), totally generating 11 orbitals

available for cluster bonding. By introducing a $d^{10}$ metal atom (Ag or Cu) to occupy one of the missing 5-connected vertices, each half of dimeric **3** and **4** attains an *arachno*-type structure. In light of this, $Ag^+$ or $Cu^+$ provides one electron for cluster bonding, leading to 22 skeleton electrons $(2n + 6)$ and fitting into $m + n + o$ rule as well. Two $Mo^0$ atoms each coordinated by three carbonyl groups occupy two 5-connected vertices, yielding a distorted monocapped square antiprism *nido*-type cluster **2**. Thereby, the skeletal electron count is consistent with the above two cases.

In order to understand the chemical bonding of clusters **1**, **2**, **3**, and **4**, we started our analysis from the *hypho*-$[Sn_2Sb_5]^{3-}$ unit. The 36 valence electrons were distributed performing adaptive natural partitioning (AdNDP) analysis[39] as implemented in AdNDP 2.0 code[40]. The AdNDP method can represent a chemical bonding pattern in terms of both Lewis bonding elements (lone pairs (1c–2e) and 2c–2e bonds) as well as delocalized bonding elements (such as nc–2e ($n > 2$) bonds), while the latter are usually associated with the concepts of aromaticity and antiaromaticity. This technique has been widely used in describing chemical bonding pattern of various Zintl clusters[41–47]. The chemical bonding of $[Sn_2Sb_5]^{3-}$ can be entirely described in terms of the classical 1c–2e and 2c–2e bonds (Supplementary Fig. 31). Specifically, seven *s*-type lone pairs with ON = 1.98–1.89 |e| were found on each Sn and Sb atoms, and one *p*-type lone pair with ON = 1.71 |e| was localized on the apex Sb-atom. The remaining 20 electrons form ten 2c–2e σ-bonds with ON = 1.99–1.92 |e|, which are responsible for the shape of the entire frame. Since the initial precursor is rather unstable in solution, it can be expected that the transition-metal coordination increases the stability of these clusters. According to the comparative chemical bonding analyses of complexes **1**–**4**, the stabilization effect can be explained by the formation of locally σ-aromatic fragments between transition metal atoms and $[Sn_2Sb_5]^{3-}$ unit.

The optimized geometry of the $[Mo_2(CO)_6Sn_2Sb_5]^{3-}$ has $C_{2v}$ symmetry, and all bond lengths are within 0.07 Å of their crystallographic counterparts. Overall, 124 valence electrons can be found as 62 two-electron bonding elements. From the optimized geometry of the cluster, it is obvious that the geometry of $[Sn_2Sb_5]^{3-}$ unit is preserved, and, hence, similar bonding fragments are anticipated in this structure. Indeed, seven 1c–2e *s*-type lone pairs and four 2c–2e Sb–Sb σ-bonds were also found in $[Mo_2(CO)_6Sn_2Sb_5]^{3-}$, similar to the $[Sn_2Sb_5]^{3-}$ cluster (Fig. 4a). However, the rest of the bonding elements were found as delocalized: four 3c–2e σ-bonds with ON = 1.91 |e| were found over the Mo–Sb–Sn units, and the remaining six electrons form three 5c–2e bonds that are responsible for the locally σ-aromatic behavior of the $Mo_2Sn_2Sb$ cap fragment. Thus, it is the formation of the multicenter 3c–2e and 5c–2e bonding elements that provides extra stabilization of the complex. In addition to AdNDP, the topology analysis of the electron localization function[48] (ELF) was performed for this cluster, as additional valuable method for determination of chemical bonding of Zintl clusters[49]. The ELF results are fully consistent with the AdNDP picture (Supplementary Fig. 39). The same bonding pattern was found for the cognate $[Cr_2(CO)_6Sn_2Sb_5]^{3-}$ complex, although with slightly different ON values (Supplementary Figs. 33–34).

A different origin of stabilization was found in the $[(AgSn_2Sb_5)_2]^{4-}$ and $[(CuSn_2Sb_5)_2]^{4-}$ clusters. As one would expect, the same bonding patterns were revealed for these species due to the valence isoelectronic Ag and Cu, in agreement with their similar geometrical structures. For the sake of simplicity, we will describe only the Cu-containing cluster below, noting that the same conclusions hold for the Ag-containing counterpart. Some of the bonds relevant to the discussion of the bonding pattern in the $[(CuSn_2Sb_5)_2]^{4-}$ cluster are shown in Fig. 4b. As in the case of the Mo-based complex, the optimized geometry of $[(CuSn_2Sb_5)_2]^{4-}$ contains two $Sn_2Sb_5$

fragments that are almost identical to the isolated $[Sn_2Sb_5]^{3-}$ unit geometry. However, in this case, the similarity is even more traced. Thus, fourteen *s*-type lone pairs with ON = 1.97–1.89 |e| and twenty 2c–2e bonds with ON = 1.98–1.80 |e| were localized, which almost completely coincides with the pattern found in $[Sn_2Sb_5]^{3-}$. Further localization leads to five doubly occupied *d*-type lone-pairs on each Cu-atom with ON = 1.99–1.97 |e|. The remaining 4 electrons form two 4c–2e Sb-Cu-Cu-Sb σ-bonds (Supplementary Fig. 36), whose combination can give two 3c–2e Cu-Cu-Sb σ-bonds (Fig. 4b). The complete bonding patterns for the Ag-containing and Cu-containing complexes are shown in Supplementary Figs. 37, 38, respectively. The shape of the $Cu_2Sb_2$ fragment, the chemical bonding picture, and the number of electrons (4e) hint at the antiaromatic character of this diamond-shaped unit. Positive $NICS_{zz}$ values at the center of $Cu_2Sb_2$ confirm the antiaromatic behavior of this unit (Supplementary Table 8). This is also in agreement with our previous studies on the $\{[CuGe_9Mes]_2\}^{4-}$ cluster featuring a similar $Cu_2Ge_2$ fragment that was found to exhibit the antiaromatic character[37]. Similarly, it is noted that the antiaromaticity of fragments with four atoms leads to the formation of two locally σ-aromatic 3c–2e islands providing stabilization (Supplementary Fig. 36). This is denoted by their negative $NICS_{zz}$ values at the center of each $M_2Sb$ triangle (Supplementary Table 8), as manifestation of the characteristics central shielding for aromatic rings, with a complementary deshielding region (positive $NICS_{zz}$ values) which overlaps at the center of the $M_2Sb_2$ diamond.

To further explore the potential global aromatic/antiaromatic characteristics of **2**–**4**, we report the induced magnetic field ($B^{ind}$) as a magnetic criterion of aromaticity (Fig. 5)[50–52]. Spherical aromatic clusters are able to sustain a long-ranged shielding cone as a distinctive property involving the overall structure. It is useful to compare directly the induced field for all four clusters despite their different shape and composition. The isotropic term ($B_{iso}^{ind}$), accounting for the average of different orientations of the applied field resembling the constant molecular tumbling in solution-state, exhibits a shielding surface for $[Mo_2(CO)_6Sn_2Sb_5]^{3-}$ owing to the presence of the multiple delocalized bonding elements (Fig. 4). It is depicted by 3c–2e and 5c–2e bonds within the cluster cage, which share common atom centers (certain atoms contribute to both multi-center bonds), resulting in a spherical-like shielding surface as observed for other Zintl-clusters and intermetalloids[41,53]. This situation is similar to the $[Ge_9]^{4-}$ case[42], where the presence of multiple σ-aromatic boding elements, sharing common atom centers, generates a spherical-like aromatic behavior given by the characteristic shielding surface (Supplementary Fig. 42). To achieve the inherent characteristics of spherical-like aromatic clusters related to the observation of a featured shielding cone, the analysis of the different terms of $B^{ind}$ accounting for specific orientations of the applied field is given. This behavior builds a direct relation between *planar* and *spherical aromatic* compounds, which is now extended to *spherical-like aromatic* species bearing multiple aromatic circuits with common atom centers in the same cage. In the case of planar aromatic species, a shielding cone is established solely when the field is oriented perpendicularly to the ring, in contrast to its appearance under different orientations of the field in three-dimensional aromatic species[54,55].

For $[Mo_2(CO)_6Sn_2Sb_5]^{3-}$, a long-ranged shielding response with a complementary perpendicular deshielding region is established under a field along the *z*-axis ($B_z^{ext}$). Interestingly, under different orientations given by $B_x^{ext}$, and $B_y^{ext}$, the same features are retained, indicating spherical aromatic behavior in $[Mo_2(CO)_6Sn_2Sb_5]^{3-}$, which is not found in the *hypho*-$[Sn_2Sb_5]^{3-}$ unit. This observation suggests that for the reactive precursor, the inclusion of $Mo(CO)_3$ fragments achieves a more favorable situation resulting in the above depicted multiple-local aromatic circuits, combining in a spherical-like aromatic cluster, able to capture the *hypho*-$[Sn_2Sb_5]^{3-}$ motif. It

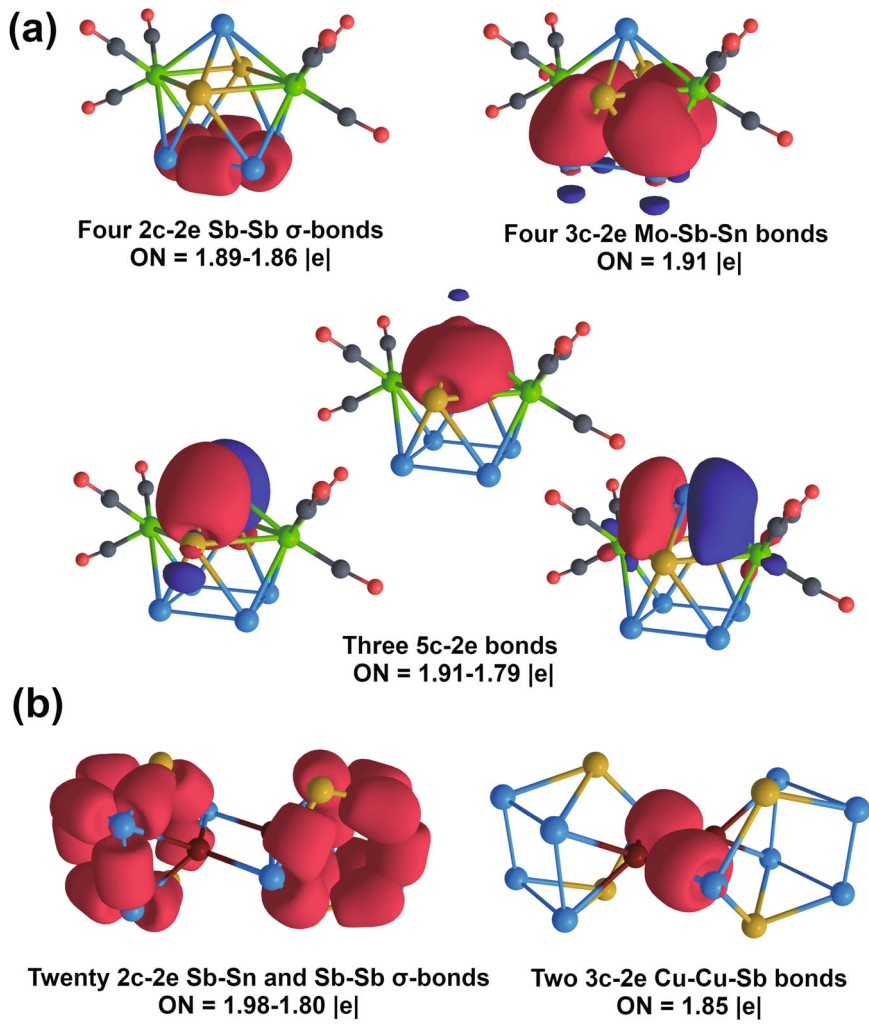

**Fig. 4 Chemical bonding pattern of the cage fragments. a** $[Mo_2(CO)_6Sn_2Sb_5]^{3-}$, **b** $[(CuSn_2Sb_5)_2]^{4-}$. ON denotes occupation numbers (equal to 2.00 |e| in an ideal case). Lines between atoms are presented for visualization and do not necessarily correspond to 2c–2e bonds. Sn-atoms are dark yellow, Sb-atoms are blue, Mo-atoms are green, C-atoms are gray, Cu-atoms are dark red and O-atoms are red.

has been recently discussed by Solà and Teixidor on the basis of *closo*-$C_2B_{10}H_{10}$, that the spherical aromaticity in deltahedral clusters for a *closo*-cluster is able to be retained after the removal of a single vertex leading to an aromatic *nido*-counterpart[56]. Our results suggest that for a *hypho*-motif, the spherical-like aromatic behavior ascribed to the *nido*-$[Mo_2(CO)_6Sn_2Sb_5]^{3-}$ is not able to be retained, as a direct consequence of the consecutive removal of cluster-vertices. Similar results are found for the Cr counterpart (Supplementary Fig. 42).

After aggregation of two *hypho*-$[Sn_2Sb_5]^{3-}$ motifs linked by two Ag(I) or Cu(I) atoms, the resulting cluster shows two separated spherical aromatic regimes as suggested by the appearance of two spherical-like shielding regions ascribed to each $MSn_2Sb_5$ side (Fig. 5b, c), from the $B_{iso}^{ind}$ term. Moreover, specific orientations of the field cause the shielding cone behavior to be exposed, which results in a dual spherical-spherical aromatic cluster with two separated spherical aromatic regimes bridged by an antiaromatic $M_2Sb_2$ diamond, similar to our findings in $\{[CuGe_9Mes]_2\}^{4-}$ with an antiaromatic $Cu_2Ge_2$ bridging unit[37]. This behavior is related to the one found for $[CB_{11}H_{11}]_2^{2-}$ with two separated spherical *closo*-carboranes[57]. The characterization of $[(AgSn_2Sb_5)_2]^{4-}$ and $[(CuSn_2Sb_5)_2]^{4-}$, retaining two spherical aromatic motifs resulting in a stable aggregate, suggests that the *hypho*-$[Sn_2Sb_5]^{3-}$ is a useful structural unit to achieve stable building blocks prone to condensate in a controlled manner.

## Discussion

In summary, we have synthesized four ternary clusters ($[M_2(CO)_6Sn_2Sb_5]^{3-}$ (M = Cr, Mo), and $[(MSn_2Sb_5)_2]^{4-}$, (M = Cu, Ag)) by the coordination of the *hypho*-$[Sn_2Sb_5]^{3-}$ to various transition metals. Through detailed theoretical analysis, we have shown that the origin of stabilization of the clusters lies in the formation of locally σ-aromatic regions that result in a spherical-like shielding surface. In this work, we have presented an effective way to lead the construction of intermediates that can be used in further studies. We believe that our findings will bring new ideas to search for unstable main-group clusters and will also contribute to our understanding of chemical bonding in hetero-metallic clusters.

## Methods

All manipulations and reactions were performed under a nitrogen atmosphere using standard Schlenk or glovebox techniques. En (Aldrich, 99%) and DMF (Aldrich, 99.8%) were freshly distilled by $CaH_2$ prior to use, and stored in $N_2$ prior to use. Tol (Aldrich, 99.8%) was distilled from sodium/benzophenone under nitrogen and stored under nitrogen. [2.2.2]-crypt (4,7,13,16,21,24-Hexaoxa-1,10-diazabicyclo (8.8.8) hexacosane, purchased from Sigma-Aldrich, 98%) was dried in a vacuum for one day prior to use. $K_8SnSb_4$ was prepared by heating a stoichiometric mixture of the elements (K: 625.6 mg, Sn: 237.4 mg, Sb: 974.4 mg; K: +99%, Sn: 99.999%, Sb: 99.9%, all from Strem) at a rate of 70 °C per hour to 700 °C and keeping it for 36 h in sealed niobium containers closed in evacuated quartz ampules according to the previous procedures[26]. The $K_8SnSb_4$ solid was obtained with a

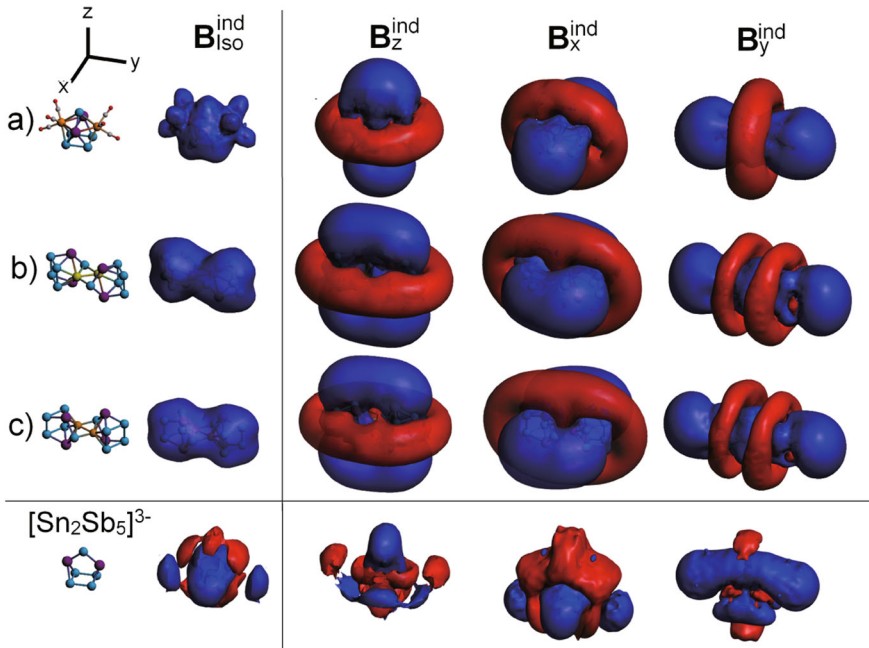

**Fig. 5 Plots of magnetic response of the studied clusters.** Induced magnetic field accounting for the orientation averaged ($B^{ind}_{iso}$) term, and for different orientation of the external field along three representative axes, for **a** $[Mo_2(CO)_6Sn_2Sb_5]^{3-}$, **b** $[(AgSn_2Sb_5)_2]^{4-}$, **c** $[(CuSn_2Sb_5)_2]^{4-}$. Blue surface: Shielding; Red surface: Deshielding. Isovalue set to ±2 ppm. Sn-atoms are dark purple, Sb-atoms are blue, Mo-atoms are orange, Ag-atoms are green, Cu-atoms are dark yellow, C-atoms are gray, and O-atoms are red.

high yield (~92%, 1.7 g) and stored under a dry nitrogen atmosphere in a glove box. $Cr(CO)_6$ and $Mo(CO)_6$ were purchased from Aldrich while $Ag_4Mes_4$ and $Cu(PPh_3)Cl$ were synthesized according to the literature with a yield of 55 and 60%, respectively[58,59].

**Synthesis of [K(2.2.2-crypt)]₃[Cr₂(CO)₆Sn₂Sb₅] (1′).** $K_8SnSb_4$ (50 mg, 0.055 mmol) and 2.2.2-crypt (83 mg, 0.220 mmol) were dissolved in 2.5 mL en and stirred for 0.5 h to yield a dark brown solution. $Cr(CO)_6$ (25 mg, 0.110 mmol) was added into the reaction mixture resulting in a red-brown suspension, and stirred for 2.5 h at 60 °C. The resulting solution was filtered through glass wool and transferred to a test tube, then carefully layered by toluene (3 mL) to allow for crystallization. After one week, dark red plate crystals of [K(2.2.2-crypt)]₃[Cr₂(CO)₆Sn₂Sb₅] (1′) in approximately 15% yield (based on $Cr(CO)_6$). [K(2.2.2-crypt)]₂[Sn₂Sb₂] as a side product was also crystallized on the wall of the test tube.

**Synthesis of [K(2.2.2-crypt)]₃[Mo₂(CO)₆Sn₂Sb₅] (2′).** $K_8SnSb_4$ (50 mg, 0.055 mmol) and 2.2.2-crypt (83 mg, 0.220 mmol) were dissolved in 2.5 mL en and stirred for 0.5 h to yield a dark brown solution. $Mo(CO)_6$ (33 mg, 0.110 mmol) was added into the reaction mixture resulting in a red-brown suspension, and stirred for 2.5 h at 60 °C. The resulting solution was filtered through glass wool and transferred to a test tube, then carefully layered by toluene (3 mL) to allow for crystallization. After 1 week, dark red plate crystals of [K(2.2.2-crypt)]₃[Mo₂(CO)₆Sn₂Sb₅] (2′) in ~20% yield (based on $Mo(CO)_6$) and [K(2.2.2-crypt)]₂[Sn₂Sb₂] crystallized on the wall of the test tube.

**Synthesis of [K(2.2.2-crypt)]₄[(AgSn₂Sb₅)₂] (3′).** In vial 1, $K_8SnSb_4$ (50 mg, 0.055 mmol) and 2.2.2-crypt (83 mg, 0.220 mmol) were dissolved in 2.5 mL en and stirred for 0.5 h to yield a dark brown solution. In another vial, $Ag_4(Mes)_4$ (25 mg, 0.027 mmol) was dissolved in 0.5 mL tol resulting in a white suspension, and added into vial 1 dropwise. The reaction mixture turned into yellow-brown and stirred for 3.5 h at room temperature. The resulting solution was filtered through glass wool and transferred to a test tube, then carefully layered by toluene (3 mL) to allow for crystallization. After five months, brown thin plate crystals of [K(2.2.2-crypt)]₄[(AgSn₂Sb₅)₂] (3′) was isolated in 12% (based on $Ag_4(Mes)_4$) together with red crystals [K(2.2.2-crypt)]₂[Sn₇Sb₂]. The crystals could also be isolated in one month but the crystal data quality was usually not as good as from long reaction time.

**Synthesis of [K(2.2.2-crypt)]₄[(CuSn₂Sb₅)₂] (4′).** $K_8SnSb_4$ (70 mg, 0.075 mmol) and 2.2.2-crypt (112 mg, 0.300 mmol) were dissolved in 2.5 mL en and stirred for 0.5 h to yield a dark green solution. $Cu(PPh_3)Cl$ (27 mg, 0.075 mmol) was added into the reaction mixture resulting in a yellow-brown suspension. After stirring for 3 h, the mixture was centrifuged for 10 min at 8000 r/min. The supernatant was filtered through glass wool and transferred to a test tube, then

carefully layered by toluene (3.5 mL) to allow for crystallization. After one week, brown block crystals of [K(2.2.2-crypt)]₄[(CuSn₂Sb₅)₂] (4′) was isolated in 15% (based on $Cu(PPh_3)Cl$) together with red crystals [K(2.2.2-crypt)]₂[Sn₇Sb₂].

**Theoretical methods.** All structures were optimized at the DFT level using the PBE0[60] hybrid density functional and def2-TZVP basis set[61]. The Gaussian 16 code[62] was used for the optimization procedures. The same level was used for chemical bonding analysis including the adaptive natural density partitioning (AdNDP)[39,40] analysis and electron localization function (ELF)[48] analysis. The AdNDP 2.0 code was used to perform AdNDP analysis, which is based on the general ideas of the NBO analysis[63]. The topology analysis of ELF was performed with the MultiWFN program[64]. The ChemCraft 1.8 software was used to visualize chemical bonding patterns and geometries of investigated compounds. The isomeric structures of $[Sn_2Sb_5]^{3-}$ were explored via an unbiased Coalescence Kick (CK) algorithm[27,28]. 2000 sample structures were optimized at PBE0/LANL2DZ[65] level. The isomeric structures of $K_3[Sn_2Sb_5]$ were explored at the same level with 4000 sample structures. The seven lowest structures for both CK calculations then were reoptimized using different functionals and def2-TZVP basis set (Supplementary Tables 6, 7). The frequency calculations were performed using the harmonic approximation, and all reported structures were confirmed to be minima due to the absence of imaginary frequencies. To assess the aromaticity and antiaromaticity with a quantitative parameter, NICS calculations were performed at PBE0/def2-TZVP level.

The nucleus-independent shielding tensors ($\sigma_{ij}$)[52,66,67] were calculated within the GIAO formalism by using the ADF2019 code[68], employing the OPBE[60,69,70] functional and an all-electron STO-TZ2P basis set, placed in a three-dimensional grid in order to evaluate the induced field ($B^{ind}$), upon an external magnetic field ($B^{ext}$), related via $B_i^{ind} = -\sigma_{ij}B_j^{ext}$[52,66,67,71–73]. For convenience, the i and j suffixes are related to the x-, y- and z-axes of the molecule-fixed Cartesian system (i, j = x, y, z). The values of $B^{ind}$ are given in ppm in relation to $B^{ext}$. Relativistic effects are considered through the ZORA Hamiltonian[74]. Solvation effects were considered via the COSMO module as implemented in the ADF code by using ethylenediamine as a solvent.

**X-ray diffraction.** Suitable single crystals of compound 1′, 2′, 3′, and 4′ were selected for X-ray diffraction analyses. Crystallographic data were collected on Rigaku XtalAB Pro MM007 DW diffractometer with graphite monochromated Cu Kα radiation (λ = 1.54184 Å). Structures were solved using direct methods and then refined using SHELXL-2014 and Olex2 to convergence[75–77], in which all the non-hydrogen atoms were refined anisotropically during the final cycles and all hydrogen atoms of the organic molecule were placed by geometrical considerations. The disorder existed in cryptand molecules of compound 3′ and heavy atoms of cluster 3 and 4 were solved by the Split SAME process in Olex 2. The disordered two toluene solvent molecules of compound 1′ and 2′ were squeezed by the Solvent

Mask in Olex 2. The specific bond angles and distances were constrained by the "Dfix" order.

**Electrospray ionization mass spectrometry (ESI-MS) investigations**. Negative ion mode ESI-MS of the DMF solutions of crystals of **1'** and **2'** were measured on an LTQ linear ion trap spectrometer by Agilent Technologies ESI-TOF-MS (6230). The spray voltage was 5.48 kV and the capillary temperature was kept at 300 °C. The capillary voltage was 30 V. The samples were made up in a glovebox under a nitrogen atmosphere and rapidly transferred to the spectrometer in an airtight syringe by direct infusion with a Harvard syringe pump at 0.2 mL/min.

**Energy dispersive X-ray (EDX) spectroscopic analysis**. EDX analysis on the compounds of **1'**, **2'**, **3'**, and **4'** was performed using a scanning electron microscope (FE-SEM, JEOL JSM-7800F, Japan). Data acquisition was performed with an acceleration voltage of 15 kV and an accumulation time of 60 s.

## Data availability

The data that support the findings of this study are available from the corresponding authors on a reasonable request. The X-ray crystallographic of compounds 1, 2, 3, and 4 reported in this study have been deposited at the Cambridge Crystallographic Data Centre (CCDC) under deposition numbers 2080455 and 2080457–2080459. These data can be obtained free of charge from The Cambridge Crystallographic Data Centre via www.ccdc.cam.ac.uk/data_request/cif.

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

## Acknowledgements

This work was supported by the National Natural Science Foundation of China (21971118 to Z.M.S.) and the USA National Science Foundation (grant CHEM-1664379) to A.I.B. A.I.B. also acknowledges financial support from the R. Gaurth Hansen Professorship fund. The work was also supported by the US Department of Energy through the Los Alamos National Laboratory. Los Alamos National Laboratory is operated by Triad National Security, LLC, for the National Nuclear Security Administration of the U. S. Department of Energy (Contract No. 89233218CNA000001). I.A.P. acknowledges support from J. Robert Oppenheimer Distinguished Postdoctoral Fellowship at the Los Alamos National Laboratory. The support and resources from the Centre for High Performance Computing at the University of Utah are gratefully acknowledged. A.M.C. acknowledges financial support from FONDECYT 1180683. We also thank the valuable discussion with John E. McGrady (Department of Chemistry, University of Oxford).

## Author contributions

Z.M.S. conceived the project and designed the experiments. Y.H.X. performed the synthesis. N.V.T., I.A.P., A.M.C., and A.I.B. performed the quantum chemical calculations and analyzed the data. L.Q. performed the single-crystal X-ray diffraction and analyzed the data. All authors co-wrote the manuscript.

## Competing interests

The authors declare no competing interests.
