## [Peer Review File · Nature Communications]

REVIEWER COMMENTS

Reviewer #1 (Remarks to the Author):

Zhong-Ming Sun et al. present four cluster compounds that they describe as hybrids of carbonyl complexes and classical Zintl polyanions. The compounds are new and probably interesting for the community that works in the field. The concept to substitute a main group element in a polyanion by a metal carbonyl fragment is not new and has been exhaustively exploited (e.g. Sevov et al. Eichhorn et al., Faessler et al., Dehnen et al.). Overall, the manuscript is rather detailed and more suited for a specialized journal such as Inorg. Chem. For what journal ever, the manuscript needs major revision with respect to the following points:

- (1) The authors mix two contradictory concepts in their description and wording: The Zintl concept fully relies on 2c2e single bonds. In contrast, the Wade-Mingos rules provide empirical electron counting rules for clusters with multicenter bonding. For several polyhedral polyanions and polycations, both concepts can formally be used, but the arrangement of atoms in such polyhedra can typically not be explained by the orientation of atomic orbitals in the 2c bonding approach, which makes the Zintl approach highly questionable. Moreover, various quantum-chemical bonding analyses have demonstrated the multicenter bonding in such polyhedra. Thus I strongly recommend to delete the Zintl terminology in this manuscript.
- (2) The idea of closo, nido and arachno polyhedra for the explanation of clusters of the heavy p-block elements was not introduced by Dehnen et. al, but decades before by various scientists, based on electron counts first derived from qualitative, later calculated MO diagrams.
- (3) Parts of the discussion of the clusters is highly misleading. The bonding orbitals comprise the entire cluster, thus, formally cutting the unit $[\text{Sn}_2\text{Sb}_5]^{3-}$ out of them and discussing them as independent hypohydro-species is inappropriate (and could be done with every borane-analogous cage). In fact, it is a substitution of vertices, in analogy to heteroatoms in a carborane or a metallaborane.
- (4) The "dimer of two arachno-cages $(\text{MSn}_2\text{Sb}_5)_2^{2-}$ sharing an M_2Sb_2 diamond" should be discussed as a conjuncto-cluster.

Moreover, it could be worth considering the positions of the "lost" vertices in arachno cages, which are adjacent e.g. in boranes but opposite e.g. in bismuth polycations.

(5) Concerning spherical aromaticity, there is a contradictory paper by Kusnetzov et al (Crystals 2020) which should be considered.

(6) Experimental weaknesses:

- (a) "After five months, brown thin plate crystals of ... (3') was isolated in 12% together with red crystals ..." does not sound like a reproducible synthesis protocol.
- (b) What is an "invalid reflection"? Any manipulation of experimental data needs a clear and detailed explanation!
- (c) All structure determinations have reliability indices wR_2 larger than 20 % and high residual electron densities, which is, despite the disorder in the organic part, very hard to understand in view of data sets that reach up to diffraction angles of 147° . Maybe the "invalid" reflections could be key to a crystallographic issue.

Reviewer #2 (Remarks to the Author):

Comments to authors

1. What are the major claims of the paper?

Zintl anions are important structural units in organic chemistry. New discovery of zintl-like anions is surely attractive. The work synthesized four novel clusters comprising a new and interesting unit $[\text{Sn}_2\text{Sb}_5]^{3-}$. By comparing the DFT and synthetic structures, the anion show very stable features, consistent with the well-retained structure of $[\text{Sn}_2\text{Sb}_5]^{3-}$ in the four compounds. Their very detailed electronic analysis support the stability of clusters. The unexpected finding of the stable $[\text{Sn}_2\text{Sb}_5]^{3-}$ would inspire future synthesis of more complicated zintl structures. Moreover, the derived constructive strategies based on $[\text{Sn}_2\text{Sb}_5]^{3-}$ can

help design novel clusters with tailored properties

2. Are they novel and will they be of interest to others in the community and the wider field?

Yes.

3. Is the work convincing?

Yes.

4. On a more subjective note, do you feel that the paper will influence thinking in the field?

Yes.

In general, constructing new structures based on a simple unit is not straightforward, since the stability of the resulted compound highly depends on the stability of the unit itself and the assembly methods. By showing several novel synthetic compounds, the present work demonstrated new and nice examples in this regard.

Suggestions for revision:

1. On page 4, the work used "the potential energy surface of $[\text{Sn}_2\text{Sb}_5]^{3-}$ was explored via the unbiased coalescence kick". I suggest to replace "potential energy surface" by "isomeric structures" since only isomeric structures were presented and no transition state were included.

2. The DFT studies of $[\text{Sn}_2\text{Sb}_5]^{3-}$ without single-point energy calculations at correlational level predicted two lower-lying isomers in C₂ and C₁-symmetries than the desired C_{2v}. In principle, the model study simply at DFT level for highly charged species would always experience such awkward cases, in comparison with the experimentally observed results. The mismatch between theory and experiment must be: 1) lack of single-point correlational energy calculations, 2) the lack of counterions. I think that both factors or either one should place C_{2v} structure to be lower in energy. These points should be commented in the revised form.

3. The theoretical analysis is very thorough and detailed. Yet I find the materials too much to be placed in the context. I suggest to move some parts to the supporting information. Only the key results put in the text is OK.

Reviewer #3 (Remarks to the Author):

See attached file [provided below Reviewer #4's comments]

Reviewer #4 (Remarks to the Author):

In this work, it is reported the synthesis and structural characterization of the systems $[\text{M}_2(\text{CO})_6\text{Sn}_2\text{Sb}_5]^{3-}$ (M = Cr, Mo), and $[(\text{MSn}_2\text{Sb}_5)_2]^{4-}$ (M = Cu, Ag). Interestingly, the 3D exotic "hypo" $[\text{Sn}_2\text{Sb}_5]^{3-}$ units function as assembly units in the $(\text{MSn}_2\text{Sb}_5)_2^{4-}$ dimers, which provides the possibility to generate nanostructures with control of their properties. The research is fascinating, and the results are outstanding because it allows the completion of the unknown chemistry of Zintl-type compounds. I am not an expert in the experimental part, but I consider that the reported results are precise and allow the reported systems' structural elucidation. Overall, I consider that these research results are essential and deserve to be published in Nature Communications. However, in my opinion, in its present state, the paper is written for a more specialized audience, it is difficult to establish the work's contribution without a thorough review. Perhaps Fig. 1 could be revised to establish the rules of this work's design and achievement (including the identification of 1-4 compounds).

Also, the discussion of aromaticity in the text is confusing, as it states: compounds 1 and 2 (monomers) are aromatic, and compounds 3 and 4 (dimers or assemblies) are antiaromatic. However, this antiaromaticity of the latter arises in the square lattice formed by the monomers upon bonding. Moreover, in the magnetic response analysis, it is established that the aromaticity of 1 and 2 (3D) persists in the dimers. The AdNDP analysis does not detect (or has not been considered) 3D aromaticity but only detects antiaromaticity (2D) in the Sb_2M_2 connection. However, despite this local antiaromaticity (of the dimers),

local 3D aromaticity (now) persists in the assemblies. In addition to the NICS at the centers of the Sb₂M₂ fragments, it would be interesting to show the Bindz planes. It is also confusing to talk about two aromatic islands giving rise to an antiaromaticity of the Sb₂M₂ fragment. Since there is delocalization (3c-2e), it should be local aromaticity. The manifestation of the paratropic region in the center of the ring (detected by NICS) is a consequence of the characteristic paratropic zone around every aromatic ring; in this case, it would be the Sb₂M moiety.

Also, there are some details in the calculations that should be checked. In the case of the potential energy surface exploration, there is an error in the sampling performed. As mentioned in the article: *Inorg. Chem.* 2019, 58, 15, 10057-10064, the cage-like structures of Zintl-type clusters with a high negative charge are not energetically preferred, given the high charge tends to the repulsive interactions are high and dissociative structures tend to be preferred. A simple strategy suggested in this work is to use simple counterions (e.g., alkali cations) or simulate a polar solvent by a continuous polarizable model. In the case of Sn₂Sb₅³⁻, the structure I have identified by a genetic method is dissociated and is 70 kcal/mol more stable than the one reported in Table S6, at the PBE0/Def2TZVP level (see coordinates at the end). In the case of the Bind analysis, there is an error in the definitions. Bindz, BindX, and BindY correspond to the magnetic field modules induced by an external field in the Z, X, and Y directions. However, there is no isotropic Bind. To maintain this format, without making conceptual errors, I suggest reporting the results as isosurfaces of NICS_{zz}, NICS_{yyy}, and NICS_{xxx}.

Other minor details:

Two versions of Gaussian are cited, 09 in the draft and 16 in the supporting material. Justify the change of program (to ADF) and level of theory for the Bind analysis, since shielding (which is where the Bind is estimated from) can be calculated with Gaussian. Change "level of theory" to level.

Cartesian Coordinates of the lowest energy isomer of Sn₂Sb₅³⁻ at PBE0/Def2TZVP level.

Sn -77.9498 -6.5879 -0.0941

Sb 43.5704 -10.5024 0.0641

Sb 45.5633 -9.2342 -1.3607

Sb 46.2648 -9.3378 1.2968

Sb 7.5749 20.516 -0.9399

Sb -75.4207 -6.1046 0.1207

Sn 9.0462 21.5442 0.9294

Review

The paper deals with the synthesis, the crystallographic and quantum chemical characterization of four new heteroatomic Zintl anion complexes. They provide strong circumstantial evidence for existence of the *hypho*-Zintl-cluster $[\text{Sn}_2\text{Sb}_5]^{3-}$ which they were unable to isolate. The authors provide a description of chemical bonding in Zintl ions with local aromatic regions in which the bonding electrons are delocalized as the stabilizing effect. This is demonstrated by ELF-Calculations and theoretical considerations. The context of the abstract as well as the introduction/main and the conclusion is clear. The figures are well designed and comprehensible. The paper features cites all the relevant literature and appropriate references, but some numerical mistakes occur in the citations. The authors should check this one more time (e.g. in text 30, but in the reference section 32).

In my opinion the paper is interesting for the specific chemical field as well as in close related disciplines, but from a crystallographic point of view some crucial checks and an appropriate documentation are missing. It may be that the quality of the data does not meet the standards for a publication in *Nature Communications*.

Why are crystallographic data in the supporting information incomplete? (missing crystal sizes, I/σ , R_{int} and absorption correction type in the table with the crystallographic data)

Did the authors collected a powder diffractogram of their “ K_8SnSb_4 ” solid?

Why are important data left out in the .cif-files?

- The authors do not give the crystal sizes for the compounds 1 and 2 as well as their transmission factors.
- The literature of the absorption correction of complex 3 as well as the transmission factors haven't been given.
- The transmission factors of complex 4 are missing.

Why are the most I/σ and R_{int} values quite poor, although a high-end “Rigaku XtalAB Pro MM007 DW” diffractometer was used? (e.g. bad crystal quality?)

Why did the authors squeeze 96 electrons away in the structure of complex 1 without mentioning it? Did they check the structure of complex 1 for their layering solvent toluene (there should be two solvent molecules)?

Why did the authors squeezed 169 electrons in the structure of complex 2 without mentioning it? Did they check the structure of complex 2 for their layering solvent toluene or a cluster disorder?

It seems the authors calculated the B^{ind} values and therefore their shielding/deshielding regions as criteria for aromaticity. Why did they not run an NMR experiment or a similiar experimental set-up? The Dehnen group published a suitable system for tin-antimony clusters using ^{119}Sn as an NMR probe.

Minor improvements:

-
- Page 1:
 - o Z12: Check the blank after “Ingenieria”
- Page 2:
 - o Z11: Change „Main“ to „Introduction”

- Page 3:
 - o Z4: $[\text{Sn}_5\text{Sb}_3]^{3-}$ was also isolated in $[\text{nBu}_4\text{P}]_3[\text{Sn}_5\text{Sb}_3]^{20-21}$
 - o Z18: Change „these“ to „the“
- Page 4:
 - o Z6: Change „reaction“ to „reactions“
 - o Z11: Check double blank between “The” and “compounds”
 - o Z13: Change „mes“ to „Mes“
 - o Z14: Check double blank between “clusters” and “while”
- Page 5:
 - o Z3: Change „C_{2v}“ to „C_{2v}“
 - o Z6: Change „not in the“ to „not in“
 - o Z8: Change „C_{2v}“ to „C_{2v}“
- Page 6:
 - o Figure 2a): Change „Cr⁰/Mo⁰“ to „Cr(CO)₆/Mo(CO)₆“ and „Ag¹/Cu¹“ to „Ag₄(Mes)₄/Cu(PPh₃)Cl“
- Page 7:
 - o Z9: Change „an“ to „a“
- Page 8:
 - o Z7: Change „for“ to „as“
- Page 9:
 - o Z1: Change „crystallographically determined range“ to „atomic“
- Page 10:
 - o Z15: Change „analyses“ to „analysis“
- Page 11:
 - o Z7: Change „C_{2v} symmetry“ to „C_{2v}-symmetry“
- Page 16:
 - o Z3: Change „after removal“ to „after the removal“
 - o Z22: Change „Discussion and conclusion“ to „Conclusion“
- Page 17:
 - o Z22: Change „an“ to „a“
- Page 18:
 - o Z12: Add „°C“ after „60“
- Page 19:
 - o Z12: Change „supernate“ to „supernatant“
 - o Z22: Change „reputed“ to „reported“
- Page 20:
 - o Z3: Change „ε“ to „ε_r“
- Page 24:
 - o Z11: Check the blank after “C.-C.”
 - o Z17: Check the blank after “Sun”
- Experimental/Synthesis section: add missing non-breaking spaces between values and units
- Check citations (e.g. Citation in text is 30, but in the reference section 32)

Thank you very much for the reviewers' comments – we very much appreciate their considered views, and our response to each is presented below. We have made quite substantial changes to the manuscript in light of these comments, and we hope that the reviewers find the narrative improved as a result. The significant changes are highlighted in yellow.

Reviewer #1 (Remarks to the Author):

Zhong-Ming Sun et al. present four cluster compounds that they describe as hybrids of carbonyl complexes and classical Zintl polyanions. The compounds are new and probably interesting for the community that works in the field. The concept to substitute a main group element in a polyanion by a metal carbonyl fragment is not new and has been exhaustively exploited (e.g. Sevov et al. Eichhorn et al., Faessler et al., Dehnen et al.). Overall, the manuscript is rather detailed and more suited for a specialized journal such as *Inorg. Chem.*

RESPONSE: We thank the reviewer for the comments on this series of cluster compounds as new and interesting results, but we respectfully disagree that the main concept in this manuscript is “to substitute a main group element in a polyanion by a metal carbonyl fragment”. As the title and the abstract of this manuscript state, we have obtained four new cluster compounds all derived from a unique building unit ‘*hypho*’ [Sn₂Sb₅]³⁻. Such ‘*hypho*’ species is elusive in Zintl chemistry, the only reported *hypho* anion is [Sn₃Bi₃]⁵⁻ which is not used for further reaction till now (*ChemistryOpen* **2016**, *5*, 306 – 310). And the *hypho*-[Sn₂Sb₅]³⁻ has never been captured or known, not to mention the functionalization of the *hypho*-[Sn₂Sb₅]³⁻ anion (cluster anions **1** and **2** here). In contrast, an *arachno* type cluster [Sn₅Sb₃]³⁻ without any functionalization or substitution was published on *Angew. Chem. Int. Ed.* **2020**, *59*, 14251–14255. It presented an unprecedented 8-vertex topology by removing two discrete 4-connected and 5-connected vertices from the 10-vertex *closo* deltahedron which filled a gap in this series of known compounds. Cluster anions **3** and **4** also represent a new type of structure of a dimer of two *arachno* cages (MSn₂Sb₅)²⁻ (M= Cu/Ag) by sharing a M₂Sb₂ diamond plane. In our work, it can be proved that [Sn₂Sb₅]³⁻ may coordinate to more transition metals and serve as a promising versatile building block. Furthermore, theoretical calculations on these four clusters revealed that aromaticity and anti-aromaticity locally exist in the two kinds of topologies respectively which provide new research models for metal-metal bonding.

Therefore, we think that our work does not simply focus on substitution of vertices. The reported vertices substitution clusters are easily to recognize from the parent clusters, for example, [(η⁵-Pb₉)Mo(CO)₃]⁴⁺ and [Ge₉Zn(C₆H₅)]³⁻ (*Eur. J. Inorg. Chem.* **2005**, 3663–3669; *Organometallics* **2006**, *25*, 4530-4536) which could be described as replacing one 5-connected and 4-connected vertex of *closo* Pb₁₀²⁻ or Ge₁₀²⁻ with transition metal fragment Mo(CO)₃ or Zn(C₆H₅). As for our work, all four clusters contain a *hypho* unit [Sn₂Sb₅]³⁻, which coordinates to different transition metals, resulting in two *nido* cages (**1** and **2**) and two dimers (**3** and **4**) each consisting of two *arachno* cages. In this sense, our work has shown an interesting finding of a unique ‘*hypho*’[Sn₂Sb₅]³⁻ and stabilized it in the corresponding *nido*- and *arachno*- cluster species respectively by further assembly with proper transition metals. In

turn, its good coordination ability also provides the opportunity to construct more complex new types of ternary systems.

For what journal ever, the manuscript needs major revision with respect to the following points:

(1) The authors mix two contradictory concepts in their description and wording: The Zintl concept fully relies on 2c2e single bonds. In contrast, the Wade-Mingos rules provide empirical electron counting rules for clusters with multicenter bonding. For several polyhedral polyanions and polycations, both concepts can formally be used, but the arrangement of atoms in such polyhedra can typically not be explained by the orientation of atomic orbitals in the 2c bonding approach, which makes the Zintl approach highly questionable. Moreover, various quantum-chemical bonding analyses have demonstrated the multicenter bonding in such polyhedra. Thus I strongly recommend to delete the Zintl terminology in this manuscript.

RESPONSE: Thanks for their careful review. We have corrected our terminology in some parts of the text, though we partially disagree with the reviewer's viewpoints for the following reasons. First, Zintl anion and Zintl cluster each have a well-established concept accepted by most people in the specialized field, that is "Zintl anions refer to anionic clusters of main group (semi-)metal atoms that are stable without the addition of a transition metal atom" and "Zintl clusters commonly refer to a more general definition allowing for a composition of both main group and transition metal atoms" (*Dalton Trans.*, **2018**, 47, 14861–14869). Actually, several reported interesting clusters containing both 2c-2e bonds and multicenter bonding have employed "Zintl" concept, e.g. $[\text{Bi}_9\{\text{Ru}(\text{cod})\}_2]^{3-}$ reported by Dehnen's team (*Angew.Chem. Int.Ed.* **2017**, 56,13253 –13258). Secondly, we wonder if the reviewer refers to Zintl–Klemm–Busmann concept which "addresses electron-precise structures comprising formal two-center–twoelectron (2c–2e) bonds, in which every atom achieves a full electron octet" (*Chem. Rev.* **2019**, 119, 14, 8506–8554). It is one of important bonding concepts widely used in Zintl chemistry but should not be used to define "Zintl" concept. With the development of Zintl chemistry, it also covers more diverse clusters, not just the electron-precise structures. Thirdly, according to the broad definitions mentioned above, the $[\text{Sn}_2\text{Sb}_5]^{3-}$ could belong to "Zintl anions" and those four clusters could be attributed to "Zintl clusters", so it is accepted that we use Zintl terminology to describe our clusters. And after checking again, we found that indeed several terms involving "Zintl" were not used accurately. We have modified them in the corresponding sections accordingly.

(2) The idea of closo, nido and arachno polyhedra for the explanation of clusters of the heavy p-block elements was not introduced by Dehnen et. al, but decades before by various scientists, based on electron counts first derived from qualitative, later calculated MO diagrams.

RESPONSE: We completely agree with the Reviewer. This comment itself is correct and professional, and there is no such misleading statement in our manuscript. We just emphasized that only one *hypho* cluster $[\text{Sn}_3\text{Bi}_3]^{5-}$ and one *arachno* cluster $[\text{Sn}_5\text{Sb}_3]^{3-}$ were reported before (*ChemistryOpen* **2016**, 5, 306 – 310; *Angew. Chem. Int. Ed.* **2020**, 59, 14251–14255). We haven't mentioned that such idea for the explanation of clusters was introduced by Dehnen et. al, perhaps there is some misunderstanding.

(3) Parts of the discussion of the clusters is highly misleading. The bonding orbitals comprise the entire cluster, thus, formally cutting the unit $[\text{Sn}_2\text{Sb}_5]^{3-}$ out of them and discussing them as independent *hypho*-species is inappropriate (and could be done with every borane-analogous cage). In fact, it is a substitution of vertices, in analogy to heteroatoms in a carborane or a metallaborane.

RESPONSE: We understand the Reviewer's concern, and would like to clarify that this work is not a simple substitution of vertices and reasons have been given in the first part of response. Actually, we

considered the unit $[\text{Sn}_2\text{Sb}_5]^{3-}$ as a building block (just like the $[\text{AuPb}_{11}]^{3-}$ as a building block in $[\text{Au}_8\text{Pb}_{33}]^{6-}$ and $[\text{Au}_{12}\text{Pb}_{44}]^{8-}$ which has been mentioned in this manuscript and $(\text{Ge}_x\text{As}_{4-x})^{x-}$ ($x = 2, 3$) as building blocks for $[\text{Au}_6(\text{Ge}_3\text{As})(\text{Ge}_2\text{As}_2)_3]^{3-}$ (*Angew. Chem. Int. Ed.* **2020**, *59*, 16638–16643)) to discuss its coordination towards different transition metals and we didn't cut the unit out of the entire clusters to analyze its bonding. It is unreasonable to cut a unit from a cluster as an individual to discuss its bonding, but it is appropriate to take it as a building block to deduce the formation process of the cluster. Moreover, regarding $[\text{Sn}_2\text{Sb}_5]^{3-}$ as a hypoh species is easier for readers to understand the structures of this series of clusters and the relationships among them.

It is also useful to consider $[\text{Sn}_2\text{Sb}_5]^{3-}$ unit from the chemical bonding point of view. We can observe the evolution of the chemical bonding from the unit $[\text{Sn}_2\text{Sb}_5]^{3-}$ building block to clusters **1**, **2**, **3**, and **4**. Since the $[\text{Sn}_2\text{Sb}_5]^{3-}$ species is a building block of the synthesized clusters, some of its bonding elements preserved in the transition metal coordinated complexes. Thus, in $[(\text{AgSn}_2\text{Sb}_5)_2]^{4-}$ and $[(\text{CuSn}_2\text{Sb}_5)_2]^{4-}$ clusters, the entire Sn_2Sb_5 framework is bound in the same manner as in the $[\text{Sn}_2\text{Sb}_5]^{3-}$ unit. While for $[\text{Mo}_2(\text{CO})_6\text{Sn}_2\text{Sb}_5]^{3-}$ the formation of delocalized bonding elements can be observed, indicating the different nature of the cluster stabilization. So, analyzing the chemical bonding in the building block in relation to the chemical bonding in the more complex clusters **1-4** is very important for understanding of the evolution of their bonding patterns that helps explain their properties.

(4) The “dimer of two arachno-cages $(\text{MSn}_2\text{Sb}_5)_2^{2-}$ sharing an M_2Sb_2 diamond” should be discussed as a conjuncto-cluster. Moreover, it could be worth considering the positions of the “lost” vertices in arachno cages, which are adjacent e.g. in boranes but opposite e.g. in bismuth polycations.

RESPONSE: We thank the Reviewer for the suggestion. We employed such description because both clusters consist of two identical moieties $(\text{MSn}_2\text{Sb}_5)_2^{2-}$ ($\text{M}=\text{Cu}, \text{Ag}$) by sharing a M_2Sb_2 diamond plane with antiaromaticity but not simple 2c-2e bonds. Such situation is similar to $\{[\text{CuSn}_5\text{Sb}_3]^{2-}\}_2$ (*Angew. Chem. Int. Ed.* **2016**, *55*, 1-7) in which they described this cluster as a dimer. Furthermore, the two *arachno* cages are identical and the entire clusters show a C_{2h} symmetry, so we still stick to the original wording.

Taking the $[\text{AgSn}_2\text{Sb}_5]^{2-}$ *arachno* cage as an example, the topology is related to a 10-vertex *closo*-deltahedron by removal of two discrete 4- and 5-connected vertices which is isostructural with the *arachno*-Zintl ion $[\text{Sn}_5\text{Sb}_3]^{3-}$ (*Angew. Chem. Int. Ed.* **2020**, *59*, 14251–14255). According to Wade-Mingos rules, an *arachno* cluster with n vertices has a characteristic skeletal electron count of $2n + 6$ and $[\text{AgSn}_2\text{Sb}_5]^{2-}$ has a skeletal electron count of $2 \times 2 + 3 \times 5 + 1 + 2 = 22$ which meets the rule of $2n + 6$ ($2 \times 8 + 6$). From the above, we consider such cage as *arachno* type which is different from the reported ones.

(5) Concerning spherical aromaticity, there is a contradictory paper by Kusnetzov et al (*Crystals* 2020) which should be considered.

RESPONSE: We thank the reviewer for suggesting the paper. We cited it in the main text. We want to note that our chemical bonding description does not contradict with the results obtained in the paper by Kusnetzov et al. In our work, we used an AdNDP method which produces maximally localized two-electron bonding elements. We found that synthesized clusters contain locally sigma-aromatic fragments (localized multiple-center bonding elements that agrees with the $4n + 2$ electron counting rule for each fragment). In order to confirm aromatic behavior of the clusters, we additionally performed a magnetic response analysis and found that those clusters exhibit spherically-like shielding cones. But it does not mean that our clusters are spherically aromatic. Spherically-like aromatic shielding can occur in species bearing multiple locally aromatic circuits with common atom centers in the same cage (as was

shown before in the following papers: *Angew. Chem. Int. Ed.*, 2021, 60, 9990-9995; *Nat. Commun.*, 2020, 11, 5286).

(6) Experimental weaknesses:

(a) "After five months, brown thin plate crystals of ... (3') was isolated in 12% together with red crystals ..." does not sound like a reproducible synthesis protocol.

RESPONSE: We did a large quantity of experiments to get high quality crystals of 3'. Actually, due to the epidemic situation, we were unable to enter the laboratory for nearly five months, and the samples were stored there. As a result, the crystals with good quality were obtained. Later, we also repeated this experiment. The crystals were yielded in about one month, but the quality of the crystals was not as good as that of the crystals that had been standing for five months. Such phenomenon is rather normal in Zintl chemistry field and many reported cluster compounds take several months to grow, e.g. $[\text{Cu}@\text{Pb}_9]^{3-}$ for 3 months and $[\text{Co}@\text{Sn}_6\text{Sb}_6]_{0.825}[\text{Co}_2@\text{Sn}_5\text{Sb}_7]_{0.175}^{3-}$ for 135 days (*Chem. Eur. J.* **2008**, 14, 4479 – 4483; *Angew. Chem. Int. Ed.* **2018**, 57, 15359 –15363). Furthermore, though the crystal growth time of 3' is relatively long, the growth time of the isomorphous crystal 4' is only one week which shows that this structure is stable and repeatable. We have obtained good quality crystals of 3' and it's possible to shorten the growth time by changing the reaction conditions, such as replacement of organometallic compounds.

(b) What is an "invalid reflection"? Any manipulation of experimental data needs a clear and detailed explanation!

RESPONSE: The description of "invalid reflection" appears in the statement of compound 4' in the "X-ray Diffraction" part of main text and we are delighted to report that since the initial submission we were able to grow a better specimen of 4'. The R_1 , wR_2 and R_{int} values of 4' has been reduced to 6.06%, 17.64% and 6.10% respectively, and, hence, no invalid reflections were omitted anymore.

(c) All structure determinations have reliability indices wR_2 larger than 20 % and high residual electron densities, which is, despite the disorder in the organic part, very hard to understand in view of data sets that reach up to diffraction angles of 147° . Maybe the "invalid" reflections could be key to a crystallographic issue.

RESPONSE: Since the submission, we have been able to grow and mount better specimens of compound 2' (wR_2 17.39%) and 4' (wR_2 17.64%) and the crystal data of compound 1' has been refined again. As a result, the R_1 values of all four compounds were reduced below 7.71% and the quality of other indicators was also improved significantly. We have now replaced the old crystallographic data with the new crystallographic data.

Reviewer #2 (Remarks to the Author):

Comments to authors

1. What are the major claims of the paper?

Zintl anions are important structural units in organic chemistry. New discovery of zintl-like anions is surely attractive. The work synthesized four novel clusters comprising a new and interesting unit $[\text{Sn}_2\text{Sb}_5]^{3-}$. By comparing the DFT and synthetic structures, the anion show very stable features, consistent with the well-retained structure of $[\text{Sn}_2\text{Sb}_5]^{3-}$ in the four compounds. Their very detailed electronic analysis support the stability of clusters. The unexpected finding of the stable $[\text{Sn}_2\text{Sb}_5]^{3-}$ would inspire future synthesis of more complicated zintl structures. Moreover, the derived constructive strategies based on $[\text{Sn}_2\text{Sb}_5]^{3-}$ can help design novel clusters with tailored properties

2. Are they novel and will they be of interest to others in the community and the wider field?

Yes.

3. Is the work convincing?

Yes.

4. On a more subjective note, do you feel that the paper will influence thinking in the field?

Yes.

In general, constructing new structures based on a simple unit is not straightforward, since the stability of the resulted compound highly depends on the stability of the unit itself and the assembly methods. By showing several novel synthetic compounds, the present work demonstrated new and nice examples in this regard.

RESPONSE: We thank the reviewer for the warm note of appreciation of our work.

Suggestions for revision:

1. On page 4, the work used “the potential energy surface of $[\text{Sn}_2\text{Sb}_5]^{3-}$ was explored via the unbiased coalescence kick”. I suggest to replace “potential energy surface” by “isomeric structures” since only isomeric structures were presented and no transition state were included.

RESPONSE: Thanks for their valuable advice and we have corrected the phrase in the main text accordingly.

2. The DFT studies of $[\text{Sn}_2\text{Sb}_5]^{3-}$ without single-point energy calculations at correlational level predicted two lower-lying isomers in C₂ and C₁-symmetries than the desired C_{2v}. In principle, the model study simply at DFT level for highly charged species would always experience such awkward cases, in comparison with the experimentally observed results. The mismatch between theory and experiment must be: 1) lack of single-point correlational energy calculations, 2) the lack of counterions. I think that both factors or either one should place C_{2v} structure to be lower in energy. These points should be commented in the revised form.

RESPONSE: We thank the reviewer for the valuable comments. As requested, we have included three K atoms as counterions to consider neutral species instead of highly charged ones, and performed a new search for the low-energy structures of $\text{K}_3[\text{Sn}_2\text{Sb}_5]$. We found similar low-energy geometries of Sn_2Sb_5 core in $\text{K}_3[\text{Sn}_2\text{Sb}_5]$ clusters as was observed in our initial calculations of the charged $[\text{Sn}_2\text{Sb}_5]^{3-}$ species. As the reviewer mentioned, it partially resolves the problem of the cluster ordering in the following way: the energy difference between three lowest energy structures becomes even smaller. Clearly, all three isomers are energetically degenerate, i.e. within ~1 kcal/mol at most of the DFT functionals. To address the suggestion of reviewer regarding the electron correlation, we also added the additional single point calculations at the CCSD/def2tzvp//PBE0/def2tzvp level (ZPE and optimized geometry was calculated at PBE0/def2tzvp level). All results are summarized in Table S7. In this case we also observed that energies of three lowest isomers are within ~1.7 kcal/mol, which is within the error for this type of calculations. So, it is just not possible to distinguish one isomer from another based on the thermodynamic parameter only, even at the CCSD level. However, the evident structural resemblance of the C_{2v} isomer with the available X-Ray data (Figure S43) provides additional grounds to consider this isomer as the most relevant for the discussion of the $[\text{Sn}_2\text{Sb}_5]^{3-}$ species.

3. The theoretical analysis is very thorough and detailed. Yet I find the materials too much to place in the context. I suggest to move some parts to the supporting information. Only the key results put in the text is OK.

RESPONSE: As requested, we moved the description of chemical bonding of $\text{M}(\text{CO})_3$ fragments to the

SI-file.

Reviewer #3 (Remarks to the Author):

The paper deals with the synthesis, the crystallographic and quantum chemical characterization of four new heteroatomic Zintl anion complexes. They provide strong circumstantial evidence for existence of the *hypho*-Zintl-cluster $[\text{Sn}_2\text{Sb}_5]^{3-}$ which they were unable to isolate. The authors provide a description of chemical bonding in Zintl ions with local aromatic regions in which the bonding electrons are delocalized as the stabilizing effect. This is demonstrated by ELF-Calculations and theoretical considerations. The context of the abstract as well as the introduction/main and the conclusion is clear. The figures are well designed and comprehensible. The paper features cites all the relevant literature and appropriate references, but some numerical mistakes occur in the citations. The authors should check this one more time (e.g. in text 30, but in the reference section³²).

In my opinion the paper is interesting for the specific chemical field as well as in close related disciplines, but from a crystallographic point of view some crucial checks and an appropriate documentation are missing. It may be that the quality of the data does not meet the standards for a publication in *Nature Communications*.

Why are crystallographic data in the supporting information incomplete? (missing crystal sizes, I/σ , R_{int} and absorption correction type in the table with the crystallographic data)

Did the authors collected a powder diffractogram of their “ K_8SnSb_4 ” solid?

RESPONSE: We understand the Reviewer’s concern. “ K_8SnSb_4 ” as a Zintl phase has been reported in 1988 (*Z. Naturforsch. B*, **1988**, 43, 1156 – 1160) and in 2009 was used by Dehnen’s group as a Zintl precursor. (*Chem. Eur. J.* **2009**, 15, 12968–12973). We prepared the precursor according to the previous procedures and have made several efforts to collect its powder diffractogram as the request of the reviewer. But we didn’t get a satisfied result for that the precursor was extremely unstable when it was taken out of the glove box and its crystallinity was poor. Probably the amorphous phase has been formed and there is no available signal peak in the spectrum that can be compared to those in the simulated results.

Why are important data left out in the .cif-files?

- The authors do not give the crystal sizes for the compounds 1 and 2 as well as their transmission factors.

- The literature of the absorption correction of complex 3 as well as the transmission factors haven’t been given.

- The transmission factors of complex 4 are missing.

RESPONSE: We thank the Reviewer for the advices; we have added these data in the cif-files.

Why are the most I/σ and R_{int} values quite poor, although a high-end “Rigaku XtaLAB Pro MM007DW” diffractometer was used? (e.g. bad crystal quality?)

RESPONSE: We would like to report an update with better crystallographic data. Since the initial submission, we have been able to grow and mount better specimens of compound 2’ (R_{int} 6.21%) and 4’ (R_{int} 6.10%) and the crystal data of compound 1’ has been refined again. As a result, the R_1 and R_{int} values of all four compounds were reduced below 7.71% and 7.55% and the quality of other indicators was also significantly improved. We have now replaced the old crystallographic data with the new crystallographic data.

Why did the authors squeeze 96 electrons away in the structure of complex 1 without mentioning it?

Why did the authors squeezed 169 electrons in the structure of complex 2 without mentioning it?

RESPONSE: We are sorry about the confusion. The squeezed electrons of complex 1 and 2 both come from the squeezed toluene solvent molecules by “Solvent Mask”.

Did they check the structure of complex 1 for their layering solvent toluene (there should be two solvent molecules)?

Did they check the structure of complex 2 for their layering solvent toluene or a cluster disorder?

RESPONSE: There are two toluene solvent molecules in complexes 1 and 2, respectively. But these two solvent molecules are seriously disordered, so we use the tool of “Solvent Mask” in Olex 2 to squeeze these molecules. And after collecting better crystal data, there is no cluster disorder in complex 2.

It seems the authors calculated the Bind values and therefore their shielding/deshielding regions as criteria for aromaticity. Why did they not run an NMR experiment or a similar experimental set-up?

The Dehnen group published a suitable system for tin-antimony clusters using ^{119}Sn as an NMR probe.

RESPONSE: We thank the reviewer for the suggestion. This is an interesting point as NMR of main group elements is very useful. We have carefully performed the corresponding ^{119}Sn NMR spectroscopic investigations of crystals of compounds 1' and 3' several times. However, due to their slow decomposition in the DMF solution, the results were inconclusive (in the recorded range -3000-1000 ppm for ^{119}Sn). Similar phenomenon was also occurred on $[\text{Sn}_{14}\text{Ni}(\text{CO})]^{4+}$ (*Angew. Chem. Int. Ed.* 2016, 55, 6721–6724). In addition, most products contained some impurities more or less; this further affected the test results.

Minor improvements:

-

- Page 1:

o Z12: Check the blank after “Ingenieria”

- Page 2:

o Z11: Change „Main“ to „Introduction”

- Page 3:

o Z4: $[\text{Sn}_5\text{Sb}_3]^{3-}$ was also isolated in $[\text{nBu}_4\text{P}]_3[\text{Sn}_5\text{Sb}_3]$ 20-21

o Z18: Change „these“ to „the”

- Page 4:

o Z6: Change „reaction“ to „reactions”

o Z11: Check double blank between “The” and “compounds”

o Z13: Change „mes“ to „Mes”

o Z14: Check double blank between “clusters” and “while”

- Page 5:

o Z3: Change „C2V“ to „C2v”

o Z6: Change „not in the“ to „not in”

o Z8: Change „C2V“ to „C2v”

- Page 6:

o Figure 2a): Change „Cr0/Mo0“ to „Cr(CO)6/Mo(CO)6” and „Ag1/Cu1“ to „Ag4(Mes)4/Cu(PPh3)Cl”

- Page 7:

o Z9: Change „an“ to „a”

- Page 8:

- o Z7: Change „for“ to „as”
- Page 9:
- o Z1: Change „crystallographically determined range“ to „atomic”
- Page 10:
- o Z15: Change „analyses“ to „analysis”
- Page 11:
- o Z7: Change „C2V symmetry“ to „C2v-symmetry”
- Page 16:
- o Z3: Change „after removal“ to „after the removal”
- o Z22: Change „Discussion and conclusion“ to „Conclusion”
- Page 17:
- o Z22: Change „an“ to „a”
- Page 18:
- o Z12: Add „°C“ after „60“
- Page 19:
- o Z12: Change „supernate“ to „supernatant”
- o Z22: Change „reputed“ to „reported”
- Page 20:
- o Z3: Change „er“ to „er”
- Page 24:
- o Z11: Check the blank after “C.-C.”
- o Z17: Check the blank after “Sun”
- Experimental/Synthesis section: add missing non-breaking spaces between values and units
- Check citations (e.g. Citation in text is 30, but in the reference section 32)

RESPONSE: Thanks for their careful review. We have corrected these minor mistakes.

Reviewer #4 (Remarks to the Author):

In this work, it is reported the synthesis and structural characterization of the systems $[\text{M}_2(\text{CO})_6\text{Sn}_2\text{Sb}_5]^{3-}$ ($\text{M} = \text{Cr}, \text{Mo}$), and $[(\text{MSn}_2\text{Sb}_5)_2]^{4-}$, ($\text{M} = \text{Cu}, \text{Ag}$). Interestingly, the 3D exotic "hypo" $[\text{Sn}_2\text{Sb}_5]^{3-}$ units function as assembly units in the $(\text{MSn}_2\text{Sb}_5)_2]^{4-}$ dimers, which provides the possibility to generate nanostructures with control of their properties. The research is fascinating, and the results are outstanding because it allows the completion of the unknown chemistry of Zintl-type compounds. I am not an expert in the experimental part, but I consider that the reported results are precise and allow the reported systems' structural elucidation. Overall, I consider that these research results are essential and deserve to be published in Nature Communications. However, in my opinion, in its present state, the paper is written for a more specialized audience, it is difficult to establish the work's contribution without a thorough review. Perhaps Fig. 1 could be revised to establish the rules of this work's design and achievement (including the identification of 1-4 compounds).

RESPONSE: Thanks for their valuable suggestions, we have added the context on borane in Wade Mingos rules in the first paragraph of the main text and refined the contents in Zintl field to accommodate a wider readership. Accordingly, the Fig.1 has also been revised to reflect the introduction part.

Also, the discussion of aromaticity in the text is confusing, as it states: compounds 1 and 2 (monomers)

are aromatic, and compounds **3** and **4** (dimers or assemblies) are antiaromatic. However, this antiaromaticity of the latter arises in the square lattice formed by the monomers upon bonding. Moreover, in the magnetic response analysis, it is established that the aromaticity of **1** and **2** (3D) persists in the dimers. The AdNDP analysis does not detect (or has not been considered) 3D aromaticity but only detects antiaromaticity (2D) in the Sb₂M₂ connection. However, despite this local antiaromaticity (of the dimers), local 3D aromaticity (now) persists in the assemblies. In addition to the NICS at the centers of the Sb₂M₂ fragments, it would be interesting to show the Bindz planes. It is also confusing to talk about two aromatic islands giving rise to an antiaromaticity of the Sb₂M₂ fragment. Since there is delocalization (3c-2e), it should be local aromaticity. The manifestation of the paratropic region in the center of the ring (detected by NICS) is a consequence of the characteristic paratropic zone around every aromatic ring; in this case, it would be the Sb₂M moiety.

RESPONSE: We thank the reviewer for this observation. In our electron localization analysis, we followed the main principle of the AdNDP method – get as localized bonding elements as possible. Based on this principle, for **3** and **4** we obtained a classically bonded Sn₂Sb₅ frameworks with 1c-2e and 2c-2e bonding elements, while the coordination to transition metal manifested via two 3c-2e bonds (which are locally sigma-aromatic as the reviewer mentioned). However, the shape of the Sb₂M₂ fragment, the chemical bonding picture (Figure S35), the number of electrons (4e) render this interaction as antiaromatic. In previous studies it was shown that the square geometry is unstable for antiaromatic molecules such as Li₄ (*J. Phys. Chem. A*, 2003, 107, 554). Moreover, the antiaromaticity of Li₄ leads to the formation of locally 3c-2e sigma-aromatic islands within the two Li₃ triangles. The same behavior is observed for the Sb₂M₂ fragment. We clarified the text in order to stress that the 2D-antiaromatic behavior is retained at the Sb₂M₂ fragment in dimers **3** and **4**.

In the magnetic response analysis, we observe two separated spherically-like shielding regions for each MSn₂Sb₅ unit. Although we did not observe the delocalized bonding elements in the Sn₂Sb₅ framework with AdNDP method, the delocalized nature of M₂Sb coordination can itself be a reason of such a magnetic response. Locally sigma-aromatic fragments usually looks like spherically-aromatic fragments in magnetic response analysis. The spherical-like shielding was previously observed in species bearing multiple locally aromatic circuits with common atom centers in the same cage (as was shown before in the following papers: *Angew. Chem. Int. Ed.*, 2021, 60, 9990-9995; *Nat. Commun.*, 2020, 11, 5286).

Also, there are some details in the calculations that should be checked. In the case of the potential energy surface exploration, there is an error in the sampling performed. As mentioned in the article: *Inorg. Chem.* 2019, 58, 15, 10057-10064, the cage-like structures of Zintl-type clusters with a high negative charge are not energetically preferred, given the high charge tends to the repulsive interactions are high and dissociative structures tend to be preferred. A simple strategy suggested in this work is to use simple counterions (e.g., alkali cations) or simulate a polar solvent by a continuous polarizable model. In the case of Sn₂Sb₅³⁻, the structure I have identified by a genetic method is dissociated and is 70 kcal/mol more stable than the one reported in Table S6, at the PBE0/Def2TZVP level (see coordinates at the end).

RESPONSE: We thank the reviewer for the valuable comment! We agree with the reviewer. The described tri-anion [Sn₂Sb₅]³⁻ species were not expected to be thermodynamically stable toward autodetachment of an electron, or structure dissociation. In our initial calculation, we anticipated that metastable cluster-like local minima could be located. We added an explanation to the main text. In order to justify our initial findings, we performed a new additional CK calculations for neutral K₃[Sn₂Sb₅] species. All results are summarized in the Table S7 and Table S12. We want to note that for

lower lying isomers we observe the same geometrical features of the core Sn_2Sb_5 unit. Moreover, the chemical bonding of $\text{C}_{2v}\text{-K}_3[\text{Sn}_2\text{Sb}_5]$ cluster is completely similar to the $\text{C}_{2v}\text{-}[\text{Sn}_2\text{Sb}_5]^{3-}$ (Figure S30) showing that K-atoms are just counterions, stabilizing highly negative $\text{C}_{2v}\text{-}[\text{Sn}_2\text{Sb}_5]^{3-}$ species and preventing the structure from dissociation or electron ejection. We also commented our new results in the main text.

In the case of the Bind analysis, there is an error in the definitions. Bindz, BindX, and BindY correspond to the magnetic field modules induced by an external field in the Z, X, and Y directions. However, there is no isotropic Bind. To maintain this format, without making conceptual errors, I suggest reporting the results as isosurfaces of NICSzz, NICSYYY, and NICSXXX.

RESPONSE: As denoted in the *Acc. Chem. Res.* 2012, 45, 2, 215–228 article devoted to the induced magnetic field (B^{ind}): “Note that B^{ind}_z for an external field perpendicular to the ring is equivalent to σ_{33} (33 component of the shielding tensor) or the NICS_{zz} (17, 18)”. This came from the diagonalization of the tensor where off diagonal trend to zero. In this sense the contribution from off diagonal components i.e. NICSzx, NICSzy, is negligible or small, in this sense, and in line to the current literature, B^{ind}_z is equivalent to NICSzz, in order to account for the shape and magnitude of the magnetic response under a field oriented from the z-axis. Similarly, B^{ind}_y , and B^{ind}_x , can be related to NICSyy and NICSxx isosurfaces. From this, the average of NICSxx, NICSyy and NICSzz, accounts for the average response under different orientations of the external field, which is denoted as an isotropic response, in line to solid state NMR conventions (Haeblerlen for example) in order to account for the constant motion in solution of molecules, which hinders a clear separation of x-, y- and z-axis from regular in solution-state NMR experiments. In this sense, pertinent literature is cited in the manuscript to offer the reader an historical and current background of the use of the induced magnetic field in the evaluation of aromaticity (refs 45-47).

Other minor details:

Two versions of Gaussian are cited, 09 in the draft and 16 in the supporting material. Justify the change of program (to ADF) and level of theory for the Bind analysis, since shielding (which is where the Bind is estimated from) can be calculated with Gaussian. Change "level of theory" to level.

RESPONSE: We apologize for that confusion. Only one version of Gaussian was used - Gaussian 16. We corrected that typo in the SI-file and the main text. We also changed the phrase “level of theory” to “level” throughout the manuscript, as reviewer suggested. The B_{ind} analysis was carried out on the basis of the ADF software in order to be consistent to previous works, and by its good implementation of relativistic effects at a moderate level of theory. This is given, because core electrons contribute to the induced magnetic field (*J. Phys. Chem. C* 2012, 116, 32, 17197–17203) and requires to be included in order to get its proper description.

Cartesian Coordinates of the lowest energy isomer of $\text{Sn}_2\text{Sb}_5^{3-}$ at PBE0/Def2TZVP level.

Sn -77.9498 -6.5879 -0.0941

Sb 43.5704 -10.5024 0.0641

Sb 45.5633 -9.2342 -1.3607

Sb 46.2648 -9.3378 1.2968

Sb 7.5749 20.516 -0.9399

Sb -75.4207 -6.1046 0.1207

Sn 9.0462 21.5442 0.9294

RESPONSE: As we mentioned above, the described tri-anion $[\text{Sn}_2\text{Sb}_5]^{3-}$ species were not expected to be thermodynamically stable toward autodetachment of an electron, or structure dissociation.

Additional calculations of neutral species were performed (Table S7 and Table S12). For lower lying isomers of $K_3[Sn_2Sb_5]$ we observe the same geometrical features of the core Sn_2Sb_5 unit as was in $[Sn_2Sb_5]^{3-}$ species.

REVIEWER COMMENTS

Reviewer #1 (Remarks to the Author):

The authors invested some effort to improve the experimental basis and the wording of their manuscript as well as to respond to this reviewer, which I acknowledge. I still disagree on several aspects of the interpretation for the reasons given in my first review. Nonetheless, I tend to respect this alternative view.

What I cannot accept is the misuse of Eduard Zintl's name for something that is somehow related to his work but in many aspects deviates fundamentally. Multicenter bonding and the Wade-Mingos rules were introduced about 30 years after Zintl passed away.

For a historical and scientific review, I recommend reading Reinhard Nesper's article (<https://onlinelibrary.wiley.com/doi/full/10.1002/zaac.201400403>). There the fine lines between the concepts and the struggle for a clear and consistent interpretation are outlined. Mixing it up is not at all helpful. But this relates more to the authors' rebuttal than to the manuscript.

For the scientific part, the manuscript can be published in its revised form. Still, I regard it too specialized for Nat. Commun.

Reviewer #2 (Remarks to the Author):

I again read the work with great interest. The authors have significantly improved the quality by thoroughly all the reviewers' comments. Surely, I am quite satisfied with the added computational data and the added discussions. In my mind, the manuscript not only provide several solid examples for the zintle-like clusters, they also present the combined analysis of computational and experimental evidences to show that the "old" and "mature" zintle chemistry deserve to pursued in much. It's my expectation that their discovered new clusters with unique stability would inspire future exploration both experimental and theoretical. I suggest the acceptance of the presently revised manuscript.

Reviewer #4 (Remarks to the Author):

I am satisfied with the authors' review; they took into account almost all my recommendations (when pertinent) and responded when they considered a misinterpretation in my review. Therefore, I suggest accepting this work in Nature Communications after the authors correct (consider) some minor details that I will comment on.

1) Regarding the exploration of the PES of charged species, which is now very clear, I recommend giving credit to the paper I mentioned in my revision (Inorg. Chem. 2019, 58, 15, 10057-10064).

2) I agree with the authors that Bindz corresponds to the negative of the Z component of shielding, the same for the Y and X components. However, my criticism was another, the term Bind (isotropic) has no formal definition; maybe authors could mention the isotropic NICS, which considers the X, Y, and Z components of the shielding. In short, I'm afraid I have to disagree with the use of Bind(isotropic) because it is not correct.

3) Regarding the use of ADF for calculating the Bindz, I suppose it is for convenience of calculation. The justification that relativistic effects are included considering core electrons (ZORA) and that the latter contribute to the magnetic field is a bit dangerous to argue if one considers that one of the significant criticisms of NICS (NICS_{zz} is mathematically equivalent with Bindz) is its misdiagnosis of aromaticity due

to these local effects (e.g., core contributions). In the cases studied here, I would expect that the core contribution in the center of the rings is meager, and the interpretation correctly corresponds to the (anti)aromaticity phenomenon.

Reviewer #5 (Remarks to the Author):

Dear Editor,

as requested I looked over the manuscript and the questions of referee 3 as well as the answers of the authors. Many questions raised by the referee are taken into account, improving the quality of the structural characterization of the title compounds.

However, one aspect is still missing and this is the poorly characterized starting material. Actually the answer of the authors says that they do not have any proof what the starting compound really is. Consequently, I would say that they have something with the nominal composition K_8SnSb_4 where $SnSb_4$ units are present in the solid state as outlined by Eisenmann and Klein in their original report of the Zintl phase. That the compound was already used by the Dehnen group as a Zintl-type precursor does not give further insight into the structure and composition of the precursor. This aspect in my opinion even lowers the quality of the results as this is just another reaction applying the Zintl-type precursor K_8SnSb_4 not with $ZnPh_2$ but this time with Cu, Ag, Cr or Mo reagents. These reagents and reactions are additionally frequently used by the Sun group in recent years leading to $[Cu_4@E_{18}]^{4-}$ (J. Am. Chem. Soc. 2020); $\{[CuGe_9Mes]_2\}^{4-}$ (Chem Commun. 2020); $[Ge_5Ni_2(CO)_3]^{2-}$ (Chem. Commun. 2017) etc. Hence, in the present case the transition metal reagents act as a template fishing the Sn_2Sb_5 clusters out of the dynamic solution. In case of the results of the Dehnen group also the parent nortricyclane-type anion, $[Sn_3Sb_4]^{6-}$ with a classical bonding situation exhibiting 2e2c bonds only was isolated and characterized.

In summary many of the raised questions by referee 3 are answered and only the one of the precursor is left open and thus I think the referee would be satisfied with the changes made.

However, in my opinion the paper does not fit to Nature Communications as it is just another example of the reactivity of the Zintl phase K_8SnSb_4 , leading to some new compounds with maybe some interesting bonding inside but also in this case the reported results are nothing new as similar results are already discussed by the authors in various other papers of Zintl-type clusters. Consequently, I think that the indeed nice and useful results are better placed within a specialized journal, like Dalton Transactions or Inorganic Chemistry and are not suitable for Nature Communications.

Reviewer #5 (Remarks to the Author):

Dear Editor,

as requested I looked over the manuscript and the questions of referee 3 as well as the answers of the authors. Many questions raised by the referee are taken into account, improving the quality of the structural characterization of the title compounds.

However, one aspect is still missing and this is the poorly characterized starting material. Actually the answer of the authors says that they do not have any proof what the starting compound really is. Consequently, I would say that they have something with the nominal composition K_8SnSb_4 where $SnSb_4$ units are present in the solid state as outlined by Eisenmann and Klein in their original report of the Zintl phase. That the compound was already used by the Dehnen group as a Zintl-type precursor does not give further insight into the structure and composition of the precursor. This aspect in my opinion even lowers the quality of the results as this is just another reaction applying the Zintl-type precursor K_8SnSb_4 not with $ZnPh_2$ but this time with Cu, Ag, Cr or Mo reagents. These reagents and reactions are additionally frequently used by the Sun group in recent years leading to $[Cu_4@E_{18}]^{4-}$ (J. Am. Chem. Soc. 2020); $\{[CuGe_9Mes]_2\}^{4-}$ (Chem Commun. 2020); $[Ge_5Ni_2(CO)_3]^{2-}$ (Chem. Commun. 2017) etc. Hence, in the present case the transition metal reagents act as a template fishing the Sn_2Sb_5 clusters out of the dynamic solution. In case of the results of the Dehnen group also the parent nortricyclane-type anion, $[Sn_3Sb_4]^{6-}$ with a classical bonding situation exhibiting 2e2c bonds only was isolated and characterized.

In summary many of the raised questions by referee 3 are answered and only the one of the precursor is left open and thus I think the referee would be satisfied with the changes made.

However, in my opinion the paper does not fit to Nature Communications as it is just another example of the reactivity of the Zintl phase K_8SnSb_4 , leading to some new compounds with maybe some interesting bonding inside but also in this case the reported results are nothing new as similar results are already discussed by the authors in various other papers of Zintl-type clusters. Consequently, I think that the indeed nice and useful results are better placed within a specialized journal, like Dalton Transactions or Inorganic Chemistry and are not suitable for Nature Communications.

RESPONSE: The Zintl precursor K_8SnSb_4 was synthesized using previous synthetic method reported by Eisenmann and Klein (*Z. Naturforsch. B*, 1988, **43**, 1156 – 1160). We tried to characterize its purity using powder XRD but failed. The reason is the precursor was extremely unstable when taking out of the glove box and its crystallinity was poor. It is beyond our ability to prove the purity by the help of available apparatus in our hands. It's rather common in the Zintl chemistry field that there is no valid information about the precursor characterization.

We respectfully disagree with the comments "That the compound was already used by the Dehnen group as a Zintl-type precursor does not give further insight into the structure and composition of the precursor". The significance of cluster chemistry relies not only on the synthetic process, but also their amazing structures and interesting bonding interactions. For example, reactions of an ethylenediamine solution of Zintl phase precursor K_4Ge_9 with various organometallic complexes have yielded a large number of novel clusters, including but not limited to $[Co@Ge_{10}]^{3-}$, $[Fe@Ge_{10}]^{3-}$, $[Au_3Ge_{18}]^{5-}$, $[Au_3Ge_{45}]^{9-}$, $[Ru@Ge_{12}]^{3-}$, $[Pd_2@Ge_{18}]^{4-}$,

$[\text{Cu}(\eta^4\text{-Ge}_9)(\text{PCy}_3)]^3-$, $[\text{Ge}_8\text{Fe}(\text{CO})_3]^3-$, $[(\text{Ge}_9)_2\text{In}(\text{C}_6\text{H}_5)]^4+$, $[(\text{Ni-Ni-Ni})@(\text{Ge}_9)_2]^4+$ and $[\text{Ge}_8(\text{Mo}(\text{CO})_3)_2]^4+$ (*Angew. Chem.* 2009, 121, 2032-2036; 2007, 46, 1638-1640; 2007, 46, 5310-5313; 2005, 44, 4026-4028, *J. Am. Chem. Soc.* 2009, 131, 2802-2803; 2005, 127, 7676-7677; 2014, 136, 4, 1210-1213; *Eur. J. Inorg. Chem.* 2010, 1207-1213, *Chem. Eur. J.* 2010, 16, 11145-11150, *J. Organomet. Chem.* 2012, 721-722, 53-61, *Chem. Commun.*, 2014, 50, 4181-4183). It is indisputable that we can't judge the novelty of those work simply based on their "simple" one-step synthetic method repeatedly using the same Zintl phase K_4Ge_9 as starting material. In addition, the requirements of reaction conditions are quite harsh which brings great difficulty to the growth and characterization of new crystals. The new bonding modes and conceptual advances based on these clusters have also attracted many research interests from both experimental and theoretical chemists. As for this work, the four ternary clusters with aromaticity and anti-aromaticity are sure to attract broad discussions on their bonding interactions and interesting vertices supplement structures as well.